# DiscQuant: A Quantization Method for Neural Networks Inspired by Discrepancy Theory

## Abstract

Quantizing the weights of a neural network has two steps: (1) Finding a good low bit-complexity representation for weights (which we call the quantization grid) and (2) Rounding the original weights to values in the quantization grid. In this paper, we study the problem of rounding optimally given any quantization grid. The simplest and most commonly used way to round is Round-to-Nearest (RTN). By rounding in a data-dependent way instead, one can improve the quality of the quantized model significantly.

We study the rounding problem from the lens of *discrepancy theory*, which studies how well we can round a continuous solution to a discrete solution without affecting solution quality too much. We prove that given $m = poly(1/\varepsilon)$ samples from the data distribution, we can round all but $O(m)$ model weights such that the expected approximation error of the quantized model on the true data distribution is $\leq \varepsilon$ as long as the space of gradients of the original model is approximately low rank (which we empirically validate).

Our proof, which is algorithmic, inspired a simple and practical rounding algorithm called *DiscQuant*. In our experiments, we demonstrate that DiscQuant significantly improves over the prior state-of-the-art rounding method called GPTQ and the baseline RTN over a range of benchmarks on Phi3mini-3.8B and Llama3.1-8B. For example, rounding Phi3mini-3.8B to a fixed quantization grid with 3.25 bits per parameter using DiscQuant gets 64% accuracy on the GSM8k dataset, whereas GPTQ achieves 54% and RTN achieves 31% (the original model achieves 84%).

## 1 Introduction

Modern deep learning models continue to grow in size, incurring greater challenges to train and serve these models. Post training compression methods have emerged which aim to make model inference faster and cheaper. Compressing after pretraining is desirable among practitioners who either cannot afford to train models themselves, or do not want to change the expensive training process too much. In this paper, we study post training quantization (PTQ) of the model weights. Quantization reduces the memory requirements of the model, and speeds up inference for LLMs under memory-bound settings such as the generation phase (as opposed to prefilling phase which is compute-bound) (Kwon et al., 2023).

The quantization problem can be divided into two overall steps: (1) Construct a good low bit-complexity representation for the weights (we colloquially call this the quantization grid), and (2) Round the original weights to values in the quantization grid. Within step (1), we also consider those methods which apply a transformation on the weights to better match the encoding format. There has been much recent work on weights-only PTQ for LLMs. To date, the vast majority of such research has been focused on step (1): constructing good low bit representations (Shao et al., 2024; Tseng et al., 2024a; Egiazarian et al., 2024). However, work on rounding methods is under-explored. To the best of our knowledge, Round-to-Nearest (RTN) and GPTQ (Hassibi et al., 1993; Frantar et al., 2022; 2023) are the primary

rounding methods for LLM weight quantization. RTN is a simple baseline, and GPTQ is a data dependent method which aims to match the activations of the quantized model with that of the original model layer-by-layer.

Let $f(w; s)$ be the loss function of a neural network where $w$ are original pretrained weights and $s$ is an input sample; for example $f$ can be the usual cross-entropy loss on input $s$. To find a good rounding solution, we are looking for perturbations of the original weights $w \in \mathbb{R}^n$ that correspond to values in the quantization grid, and do not increase the loss $f$ too much. We further impose the constraint that we only round each parameter up or down, this ensures that we are not changing the original model weights too much. Then the set of allowed quantization points can be pictured as vertices of a hypercube $H$ around $w$. Let $\hat{w} \in \mathbb{R}^n$ be these perturbed weights, and $\Delta f = f(\hat{w}; s) - f(w; s)$ be the resulting change in loss function for a sample $s$. We approximate $\Delta f$ via a first order Taylor expansion: $\Delta f \approx \langle \nabla_w f(w; s), \hat{w} - w \rangle$. Some prior works such as Nagel et al. (2020); Hassibi et al. (1993) assume the gradients of a pretrained model to be nearly zero, and focus on the second order terms. We show that this assumption is not always true, the average gradients are close to zero but per-sample gradients can be big; in fact the first order term is a good approximation to $\Delta f$ (see Figure 3).

Therefore, to incur a small $\Delta f$, we want $\langle \nabla_w f(w; s), \hat{w} - w \rangle \approx 0$ for $s$ sampled from the data distribution $\mathcal{D}_{\text{data}}$. Suppose we are given $m$ independent samples $s_1, s_2, \ldots, s_m \sim \mathcal{D}_{\text{data}}$, we can impose the constraints $\langle \nabla_w f(w; s), \hat{w} - w \rangle = 0$ which correspond to an affine subspace $V$ of dimension $n - m$. The intersection of the subspace $V$ and the hypercube $H$ is a convex polytope $K$. It can be shown that any vertex of $K$ should have at least $n - m$ fully rounded parameters, see Figure 1 for an illustration. Since the number of parameters $n \gg m$, any vertex of $K$ gives an almost fully rounded solution. Obviously this solution satisfies the linear constraints for the samples $s_1, s_2, \ldots, s_m$. But will it generalize to unseen samples from the data distribution $\mathcal{D}_{\text{data}}$? We prove that it can generalize if the distribution of gradients $g = \nabla_w f(w; s)$ for $s \sim \mathcal{D}_{\text{data}}$ is approximately low rank. Let $\Sigma = \mathbb{E}_{s \sim \mathcal{D}_{\text{data}}}[gg^T]$ where $g = \nabla_w f(W; s)$ be the covariance matrix of gradients. We prove the following theorem; the algorithm and the proof draws on techniques from discrepancy theory, in particular the famous Lovett-Meka algorithm (Lovett & Meka, 2012).

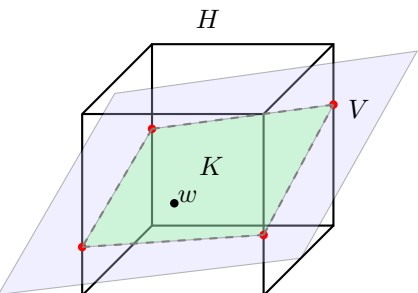

Figure 1: An illustrative figure showing the convex polytope $K$ formed by the intersection of an $n$-dimensional hypercube $H$ and an $n - m$ dimensional affine subspace $V$. Any vertex of $K$ should have $n - m$ coordinates which are fully rounded.

**Theorem 1.1** (Informal). *If the eigenvalues of the covariance matrix of gradients decay polynomially fast, then given $m = \text{poly}\left(\frac{\log n}{\varepsilon}\right)$ samples $s_1, s_2, \ldots, s_m \sim \mathcal{D}_{data}$ there is a randomized algorithm to find $\hat{w}$ with $n - m$ weights rounded such that $E_{s \sim \mathcal{D}_{data}}[|\Delta f|] \leq \varepsilon$.*

From these insights we develop a practical rounding algorithm called *DiscQuant*. The Lovett-Meka algorithm does a random walk starting from the original weights until it converges to a vertex of $K$. Instead, we can find a vertex of $K$ by minimizing a linear function over the convex polytope $K$. DiscQuant uses stochastic gradient descent to minimize two objectives, one corresponding to low $\Delta f$, and the other corresponding to minimizing a linear function. We take a knowledge distillation approach for the first term, minimizing the KL divergence between the original and quantized model. These two losses are balanced with a regularization parameter $\lambda > 0$:

$$\min_{\hat{w}} \lambda \langle c, \hat{w} \rangle + \mathbb{E}_{z \sim \mathcal{D}_{\text{data}}} \mathbb{E}_i [D_{KL}\left(p_w(\cdot | z_{<i}) \| p_{\hat{w}}(\cdot | z_{<i})\right)]$$
$$s.t. \ \hat{w} \in H.$$

$$(1)$$

Here $p_w(\cdot | z_{<i})$ is the next token distribution given prefix $z_{<i}$. An astute reader may notice that the first order approximation of the KL divergence in (1) is exactly zero, and how our

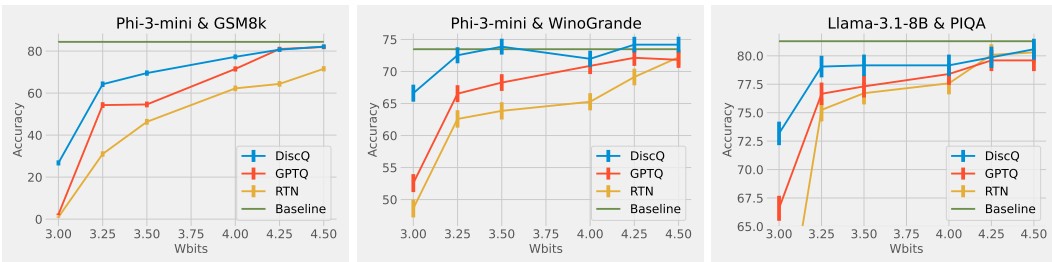

Figure 2: Select results quantizing Phi-3-mini-4k-instruct and Meta-Llama-3.1-8B-Instruct using block scaling quantization. GSM8k is a math-based generative task, and WinoGrande and PIQA are multiple choice commonsense reasoning tasks. Error bars are standard errors from lm-evaluation-harness. See Section 5 for full results.

discussion above applies. In Section 4 where we describe in detail our exact optimization objective, we also show that the second order term of KL divergence can be written as

$$\mathbb{E}_{z \sim \mathcal{D}_{\text{data}}} \mathbb{E}_i \mathbb{E}_{t \sim p_w(\cdot|z_{<i})} \left[ \langle \nabla_w \log p_w(t|z_{<i}), \hat{w} - w \rangle^2 \right].$$

So minimizing the KL divergence is a succinct way to impose constraints of the form $\langle \nabla_w \log p_w(t|z_{<i}), \hat{w} - w \rangle \approx 0$ or equivalently $\log p_w(t|z_{<i}) \approx \log p_{\hat{w}}(t|z_{<i})$ where $t \sim p_w(\cdot|z_{<i})$ and $z \sim \mathcal{D}_{\text{data}}$. Therefore our framework still applies.

After every step of gradient descent, we project the weights back to the hypercube $H$. This ensures that the trajectory of DiscQuant remains within the convex polytope $K$ and eventually converges to a vertex of $K$ with almost all the coordinates rounded. Instead of picking a random direction $c$ to find a random vertex of $K$, we use a special $c^*$ which let's us find the vertex closest to the original weights $w$ (see Section 4). We use RTN to round the few unrounded parameters left at the end of the optimization.

We perform extensive experiments which show the strength of our method: on models Phi-3-mini-4k-instruct and Meta-Llama-3.1-8B-Instruct, across a variety of evaluation tasks, and across the block scaling and incoherence processing quantization formats. DiscQuant is agnostic towards the quantization grid, and can therefore be composed with other quantization methods. Block scaling sets a `bits` parameter which determines the number of grid points, and a unique scaling parameter per `groupsize` weights (Frantar et al., 2023). Incoherence processing applies a random orthogonal transformation, which reduces the weight ranges and can make quantization easier (Chee et al., 2023; Tseng et al., 2024a). A subset of results can be found in Figure 2. Across tasks, models, and quantization levels, our method DiscQuant achieves superior compression over baselines GPTQ and RTN.

We summarize our main contributions:

- **Theoretical developments:** We prove that it is possible to achieve generalization error $\leq \varepsilon$ on the true data distribution by rounding all but $\text{poly}(\log n/\varepsilon)$ weights, so long as the gradients of the original model are approximately low rank.
- **Practical algorithm:** We develop a simple and practical algorithm DiscQuant guided by our theoretical analysis. We perform extensive experiments on Phi-3-mini-4k-instruct and Meta-Llama-3.1-8B-Instruct, over block scaling and incoherence processing quantization formats, and a variety of evaluation tasks. Our method DiscQuant achieves superior or comparable quantization to the baselines GPTQ and RTN as can be seen from Figure 2.

## 2 RELATED WORK

In this paper we focus on weights-only PTQ. Quantization can also be applied to the activations or KV-cache (Ashkboos et al., 2024; Liu et al., 2024a;b). Other compression method such as pruning (Frantar & Alistarh, 2023; Sun et al., 2023) are also outside the scope of this work. As discussed in the introduction, post training quantization can be divided into

two overall steps: (1) Construct a good low bit-complexity representations for the weights (the quantization grid), and (2) Round the original weights to the values in the quantization grid. To this date, the vast majority of PTQ research for LLMs has focused on step (1). Note that determining a good compressed representation can involve both encoding formats, as well as transformations to ensure the weights better match the encoding format.

## 2.1 QUANTIZATION GRIDS

One of the more common quantization formats is called block scaling, or group-wise quantization (Frantar et al., 2023). In addition to the `bits` parameter determining the number of representable points, each `groupsize` parameters share a unique scaling parameter. Another successful encoding is to identify a small set of important weights and keep them in high precision (Dettmers et al., 2022; 2024; Kim et al., 2024). Shao et al. (2024) learns quantization parameters. Other works apply transformations to make quantization easier, either relatively simple invariant scalings (Xiao et al., 2023; Lin et al., 2024), or more complicated random orthogonal transformations (Chee et al., 2023; Liu et al., 2024a). Beyond block scaling, there has been work quantizing multiple parameters together using vector quantization (Tseng et al., 2024a; Egiazarian et al., 2024; van Baalen et al., 2024) or trellis quantization (Tseng et al., 2024b).

## 2.2 ROUNDING

To the best of our knowledge, GPTQ (Frantar et al., 2023) is the main rounding method for LLMs. It is based on the Optimal Brain Surgeon (Hassibi et al., 1993), which was adapted for pruning and quantization in Frantar et al. (2022) and then refined for quantization in GPTQ. GPTQ works by minimizing a layer-wise objective $\|WX - \hat{W}X\|_2^2$, where $W$ is the weight matrix of a linear layer and $X$ is the matrix of input activations to that layer (stacked as columns). Two other LLM rounding methods both use coordinate descent: Nair & Suggala (2024) only has results on the closed source PaLM-2 models with no released code, and Behdin et al. (2023) has results on the OPT, BLOOM, and Falcon model families.

There was more work on rounding methods several years ago, before the LLM boom. These papers were typically on smaller vision models. The line of work was started by AdaRound (Nagel et al., 2020) and continuing to AdaQuant (Hubara et al., 2021) and BRECQ (Li et al., 2021) employ a similar approach to ours, optimizing essentially interpolation variables between the closest up($w^{\text{up}}$) and down($w^{\text{down}}$) quantization grid points, while adding a concave regularization term to encourage rounding and using a rectified sigmoid to interpolate between $w^{\text{up}}$ and $w^{\text{down}}$. They also do rounding layer by layer. However our method uses a linear term as a regularizer inspired from our theoretical insights using discrepancy theory and uses simple linear interpolation between $w^{\text{up}}$ and $w^{\text{down}}$ and we round the entire model at once.

## 2.3 DISCREPANCY THEORY

Discrepancy theory is a deep branch of mathematics and theoretical computer science, and we refer the readers to standard textbooks for more details (Matousek, 2009; Chazelle et al., 2004; Bansal, 2022) To our knowledge, only Lybrand & Saab (2021) makes the connection between discrepancy theory and quantization. However, besides the high level motivational similarities, their work is not directly relevant to ours. Lybrand & Saab (2021) reduce the problem of understanding the error introduced by quantization on the output of a single neuron to a problem in discrepancy, and construct an algorithm for quantizing a single neuron. Their theoretical analysis on the generalization error only applies to quantizing the first layer of a neural network. On the other hand, we use discrepancy theory to understand when the whole network $f(w; s)$ can be approximated by $f(\hat{w}; s)$ with $\hat{w}$ in the quantization grid, and our theory holds for any network as a whole as long as our assumptions are true.

## 3 CONNECTIONS TO DISCREPANCY THEORY

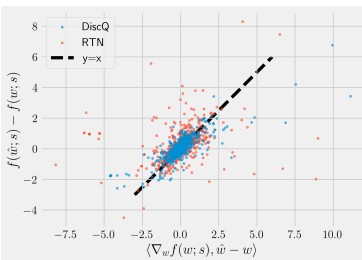
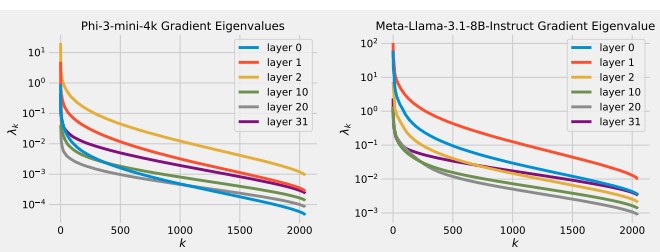

Figure 3: First order approximation of the error function $\Delta f$ when quantizing the model to 4.25 bits using RTN and Disc-Quant. Here $f$ is the per-token loss function and $s$ is sampled from the WikiText-2 dataset.

Figure 4: Eigenvalues of the covariance matrix of the gradients of pre-trained models. The covariance matrix is estimated by averaging over $8k$ sample gradients from RedPajama-1T-Sample and projecting them to 2048 dimensions using Johnson-Lindenstrauss projections.

Let $f(w; s)$ be the loss function of a pre-trained neural network with weights $w \in \mathbb{R}^n$ on an input sample $s$ and let $\mathcal{D}_{\text{data}}$ be the sample data distribution. Suppose we are also given a (scalar) quantization grid $\mathcal{Q} = Q_1 \times Q_2 \times \cdots \times Q_n$ where $Q_j \subset \mathbb{R}$ is a finite set of quantization points available to quantize the $j^{th}$ parameter.[1] In this work, we focus on scalar quantization which allows us to write the quantization grid as a product set, i.e., each parameter can be independently rounded to

| Model | $\|\mathbb{E}(g)\|^2$ | $\mathbb{E}\|g\|^2$ |
|---|---|---|
| Phi3-mini-128k | 0.1021 | 4.7812 |
| Llama3.1-8B | 1.6328 | 107 |

Table 1: $\|\mathbb{E}(g)\|^2$ vs $\mathbb{E}\|g\|^2$ over 8192 samples from RedPajama-1T-Sample dataset with window size 2048.

a finite set of available values. Alternatively, in vector quantization a group of $d$ variables are rounded together to one of a finite set of quantization points in $\mathbb{R}^d$, which has been used in some prior works (Tseng et al., 2024a; Egiazarian et al., 2024; van Baalen et al., 2024). Generalizing our method to vector quantizers is an interesting future research direction.

Our goal is to find a rounding $\hat{w} \in \mathcal{Q}$ of the original weights $w$ such $f(\hat{w}; s) \approx f(w; s)$ where $s \sim \mathcal{D}_{\text{data}}$. We further impose the constraint that for each parameter $w_j$, we only round up or round down to the available values in $Q_j$, i.e., we only have two choices for $\hat{w}_j$ denoted by $w_j^{\text{up}}, w_j^{\text{down}} \in Q_j$ where $w_j^{\text{up}} \leq w_j \leq w_j^{\text{down}}$.[2] We make this assumption because we don't want to change any parameter of the original model too much during quantization, consider it an important property of algorithms we design. Using Taylor expansion:

$$\Delta f = f(\hat{w}; s) - f(w; s) = \langle \nabla_w f(w; s), \hat{w} - w \rangle + (\hat{w} - w)^T \nabla_w^2 f(w; s)(\hat{w} - w) + \cdots \quad (2)$$

Assuming that the quantization grid $\mathcal{Q}$ is fine enough and since we only round each parameter up or down, $\|\hat{w} - w\|$ is small and so we can ignore the higher order terms. We claim that the first order term is the dominant term. Prior works such as Nagel et al. (2020); Hassibi et al. (1993); LeCun et al. (1989) have assumed that the first order term can be assumed to be zero because the model is trained to convergence and focused on reducing the second order term. But the model being trained to convergence just means that average gradient over many samples from the distribution is nearly zero. But the gradients still have some variance and gradients w.r.t. individual samples from the data distribution are not approximately zero (see Table 1). Figure 3 demonstrates this by showing that the error term $\Delta f$ is well-correlated with the first order approximation $\langle \nabla_w f(w; s), \hat{w} - w \rangle$.[3]

So the goal now is to find a rounding $\hat{w}$ such that $\langle \nabla_w f(w; s), \hat{w} - w \rangle \approx 0$ for samples $s \sim \mathcal{D}_{\text{data}}$. Suppose we sample $m$ samples $s_1, s_2, \ldots, s_m \sim \mathcal{D}_{\text{data}}$ independently from the

---

[1]The quantization grid $\mathcal{Q}$ can depend on $w$, like in Block Scaling (Frantar et al., 2023). So ideally, we should write $\mathcal{Q}_w$, but we ignore the dependence to simplify notation.

[2]If $w_j < \min Q_j$ or $w_j > \max Q_j$, we just set $w_j^{\text{up}} = w_j^{\text{down}} = \min Q_j$ or $\max Q_j$ respectively.

[3]In the special case when $f$ is the KL distillation loss between the original model and quantized model, the first order term vanishes exactly. See Section 4 for why this analysis still applies.

data distribution, where $m \ll n$. We now break our task into two parts of bounding the empirical error and generalization error as follows:

**Question 3.1.** *Can we find* $\hat{w} \in \mathcal{Q}$ *(with* $\hat{w}_j \in \{w_j^{down}, w_j^{up}\}$*) such that* $\langle \nabla_w f(w; s_i), \hat{w} - w \rangle \approx 0$ *for all the samples* $s_1, \ldots, s_m$*?*

**Question 3.2.** *Once we find such a* $\hat{w}$*, will it generalize to the true data distribution, i.e., will* $\langle \nabla_w f(w; s), \hat{w} - w \rangle \approx 0$ *for* $s \sim \mathcal{D}_{data}$*? How many samples* $m$ *do we need for this?*

### 3.1 BOUNDING EMPIRICAL ERROR (QUESTION 3.1)

For simplicity, let us assume that the quantization grid is uniform and $w_i^{\text{up}} - w_i^{\text{down}} = \delta$ for all $i \in [n]$ where $\delta > 0$ is the distance between grid points. See Appendix C for how to genealize this to non-uniform grids. We will introduce new parameters $x \in [0, 1]^n$ and define $w^x = w^{\text{down}} + \delta x$. Note that $w_i^x$ interpolates between $w_i^{\text{down}}$ and $w_i^{\text{up}}$ where $w_i = w_i^{\text{down}}$ if $x_i = 0$ and $w_i = w_i^{\text{up}}$ if $x_i = 1$. Let $y \in [0, 1]^n$ be the interpolation point corresponding to the original weights, i.e., $w^y = w$. We can rewrite the linear constraints in terms of $x$ as follows:

$$\langle \nabla_w f(w; s_i), w^x - w \rangle = \langle \nabla_w f(w; s_i), w^x - w^y \rangle = \delta \langle \nabla_w f(w; s_i), x - y \rangle.$$

Let $M$ be an $m \times n$ matrix whose $i^{th}$ row is given by $\nabla_w f(w; s_i)$. Then the linear constraints can be simply written as $M(x - y) = 0$. Our goal is to find a fully integral $\hat{x} \in \{0, 1\}^n$ such that $M(\hat{x} - y) = 0$. Let $V = \{x \in \mathbb{R}^n : Mx = My\}$ which is an affine subspace of dimension $\geq n - m$. Define $K = [0, 1]^n \cap V$ as the intersection of the hypercube with this subspace. $K$ is a convex polytope and it is non-empty because $y \in K$. Therefore any vertex of $K$ should have $n - m$ integral coordinates (i.e., coordinates $j$ such that $x_j \in \{0, 1\}$).[4]

See Figure 1 for geometric intuition about why this is true. Since the number of parameters $n$ is much larger than the number of samples $m$, any vertex of $K$ is almost fully integral and exactly satisfies all the $m$ linear constraints.

Suppose we further ask for a fully integral $\hat{x}$ which approximately satisfies all the $m$ linear constraints, this precise question is answered by *discrepancy theory* which studies how to do this and relates the approximation error to properties of $M$ such as *hereditary discrepancy* (Lovász et al., 1986; Bansal, 2022). We don't explore this direction further because the almost integral $\hat{x}$—a vertex of $K$—is good enough if we apply RTN to the few remaining fractional parameters; we observe that the linear constraints are all approximately satisfied.

### 3.2 BOUNDING GENERALIZATION ERROR (QUESTION 3.2)

How do we bound the generalization error if we know that the empirical approximation error is small? If $\hat{w} - w$ is approximately orthogonal to $m$ sample gradients $\nabla_w f(w; s_i)$ for $i = 1$ to $m$, why should we expect that $\hat{w} - w$ is orthogonal to unseen gradients $\nabla_w f(w; s)$ for samples $s \sim \mathcal{D}_{\text{data}}$? This should happen only if the gradients are approximately low rank. More precisely, let

$$\Sigma = \mathbb{E}_{s \sim \mathcal{D}_{\text{data}}}[gg^T] \text{ where } g = \nabla_w f(w; s)$$

be the covariance matrix of the distribution of sample gradients and let $\lambda_1 \geq \lambda_2 \geq \cdots \geq \lambda_n$ be its eigenvalues. We observe that the eigenvalues decay very fast, see Figure 4 for empirical validation of this on some real world models. We model this by assuming that $\lambda_k \leq \lambda_1 / k^\alpha$ for $\alpha > 1$. The assumption that $\alpha > 1$ is valid since

$$\mathbb{E}_s[\|g\|^2] = \mathbb{E}_s[\text{Tr}(gg^T)] = \text{Tr}(\mathbb{E}_s[gg^T]) = \text{Tr}(\Sigma) = \sum_{i=1}^n \lambda_i.$$

It is well-known that the gradients of a pretrained model have constant norm on most samples (see Table 1 for empirical validation). Therefore $\sum_{i=1}^n \lambda_i = O(1)$ and so the the decay coefficient $\alpha$ has to be at least 1.

---

[4]This is because at a vertex, we need to have $n$ tight constraints, and $V$ imposes only $m$ tight constraints. So the remaining $n - m$ tight constraints should come from the hypercube. These are also called basic feasible solutions in linear programming.

Under this assumption, it is reasonable to expect generalization. But this is not at all obvious to find a generalizing solution. In fact, any deterministic algorithm which chooses one of the vertices of $K$ will most likely not generalize. We give a randomized rounding algorithm (see Algorithm B.2) based on the famous Lovett-Meka algorithm from discrepancy theory (Lovett & Meka, 2012) which finds a vertex of $K$ which has low generalization error. The algorithm starts at $y$ and does a random walk (Brownian motion) inside the $n-m$ dimensional subspace $V$ formed by the linear constraints imposed by the $m$ samples. Whenever it hits a face $x_i = 0$ or $x_i = 1$ of the hypercube, it fixes that variable and continues the random walk until almost all the variables are rounded.

In order to prove rigorous bounds we also need a mild assumption that the distribution of gradients is well-behaved. We use the notion by O'Donnell (2014) and say that for a parameter $\beta \geq 1$, a random vector $X \in \mathbb{R}^n$ is *$\beta$-reasonable* if

$$\mathbb{E}[\langle X, \theta \rangle^4] \leq \beta \cdot \mathbb{E}[\langle X, \theta \rangle^2]^2 \quad \forall \theta \in \mathbb{R}^n.$$

For example $X \sim \{-1, 1\}^n$ and a Gaussian $X \sim N(\mathbf{0}, \Sigma)$ are both $O(1)$-reasonable. Our main theoretical result (proved in Appendix B) is then:

**Theorem 3.3.** *Let $\alpha > 1$ and $\beta \geq 1$ be constants and let $1 \leq m \leq \frac{n}{16}$. Let $\mathcal{D}$ be a $\beta$-reasonable distribution with unknown covariance matrix $\Sigma \in \mathbb{R}^{n \times n}$ whose Eigenvalues satisfy $\lambda_k \leq \frac{\lambda_1}{k^\alpha}$ for all $k = 1, \ldots, n$. Then there is a randomized polynomial time algorithm that given a $y \in [0,1]^n$ and $m$ independent samples $g_1, \ldots, g_m \sim \mathcal{D}$, produces an $x \in [0,1]^n$ with high probability such that all but $O(m)$ parameters in $x$ are fully rounded and*

$$\mathbb{E}_{g \sim \mathcal{D}}[\langle g, x - y \rangle^2] = (x - y)^T \Sigma (x - y) \lesssim_{\alpha, \beta} \lambda_1 m^{-\min\{1/2, \alpha-1\}} (\log n)^2.$$

# 4 DiscQuant: Algorithm

In this section, we will present *DiscQuant*, a simple and practical algorithm for rounding inspired by the theoretical insights in Section 3. Instead of trying to approximate the loss function of the pre-trained model, i.e., $f(\hat{w}; s) \approx f(w; s)$, we will instead take a distillation approach and try to minimize the KL divergence between the next token distribution of the original model and the quantized model. Let $p_w(\cdot|z_{<i})$ be the distribution of the next token predicted by the original model given prefix $z_{<i}$ where $z \sim \mathcal{D}_{\text{data}}$ is a sample from the data distribution. We want $\text{error}(\hat{w}) = \mathbb{E}_{z \sim \mathcal{D}_{\text{data}}} \mathbb{E}_i D_{KL} \left( p_w(\cdot|z_{<i}) \parallel p_{\hat{w}}(\cdot|z_{<i}) \right) \approx 0$.

Expanding $\text{error}(\hat{w})$ using Taylor series, we can see that first order term vanishes exactly and so the second order term is the dominant term (see Appendix D). By Lemma D.1, Hessian of $\text{error}(\hat{w})$ can be written as a covariance of gradients as:

$$H_w = \mathbb{E}_{z \sim \mathcal{D}_{\text{data}}} \mathbb{E}_i \mathbb{E}_{t \sim p_w(t|z_{<i})} \left[ (\nabla_w \log p_w(t|z_{<i})(\nabla_w \log p_w(\cdot|z_{<i}))^T \right].$$

Therefore

$$\text{error}(\hat{w}) \approx (\hat{w} - w)^T H_w (\hat{w} - w) = \mathbb{E}_{z \sim \mathcal{D}_{\text{data}}} \mathbb{E}_i \mathbb{E}_{t \sim p_w(\cdot|z_{<i})} \left[ \langle \nabla_w \log p_w(t|z_{<i}), \hat{w} - w \rangle^2 \right].$$

So minimizing $\text{error}(\hat{w})$ is a succinct way to impose constraints of the form $\langle \nabla_w \log p_w(t|z_{<i}), \hat{w} - w \rangle \approx 0$ or equivalently $\log p_w(t|z_{<i}) \approx \log p_{\hat{w}}(t|z_{<i})$ where $t \sim p_w(\cdot|z_{<i})$ and $z \sim \mathcal{D}_{\text{data}}$. Therefore, we can use the same techniques developed in Section 3 to solve this as well. Assuming that the gradients are low rank, the set of $x$ satisfying these constraints (where $\hat{w} = w^x$) form an affine subspace $V$ of dimension $\geq n - m$ where $m$ is the number of samples. We are again interested in finding a vertex of the polytope $K = [0,1]^n \cap V$ which will have $\geq n - m$ integral coordinates. At this point, we could use the Lovett-Meka algorithm (Algorithm B.2) which has provable generalization guarantees. But explicitly calculating all the gradients and storing them is infeasible. Instead a simple heuristic way to find a random vertex of polytope $K$ is to minimize a random linear function. Let $c \in \mathbb{R}^n$ be some arbitrary vector; we will try to minimize the linear function $\langle c, x \rangle$ along with the KL divergence by taking a linear combination of them. The final optimization objective is shown in (3) where $\lambda > 0$ is a regularization coefficient.

$$\min_x \lambda \langle c, x \rangle + \mathbb{E}_{z \sim \mathcal{D}_{\text{data}}} \mathbb{E}_i [D_{KL} \left( p_w(\cdot|z_{<i}) \parallel p_{w^x}(\cdot|z_{<i}) \right)]$$
$$s.t. \ x \in [0,1]^n. \tag{3}$$

We solve the optimization problem (3) using projected stochastic gradient descent where we project $x$ to the hypercube after every gradient update. Optimizing (3) will keep us close the polytope $K$ and will approximately converge to a vertex of $K$ which is almost integral. We round whatever fractional coordinates are left using RTN to get a fully integral solution.

We use one additional heuristic to improve the performance of the algorithm in practice. Instead of choosing a random vertex of the polytope $K$ by choosing the vector $c$ at random, we will choose it carefully so as to find the vertex of the polytope $K$ which is closest to $y$ which is the interpolation point corresponding to the original model weights (i.e., $y$ such that $w^y = w$). We have:

$$\|x - y\|^2 = \sum_i (x_i^2 - 2x_i y_i + y_i^2) \approx \sum_i (x_i - 2x_i y_i + y_i^2) = \langle c^*, x \rangle + \|y\|^2$$

where $c^* = (1 - 2y)$. Here we have used the fact that $x_i^2 = x_i$ whenever $x_i \in \{0, 1\}$ and since $x$ is almost integral, we can use the approximation in the summation above. With this approximation, minimizing $\|x - y\|^2$ over almost integral $x$ is equivalent to minimizing $\langle c^*, x \rangle$. So in the DiscQuant algorithm, we use $c = c^*$ specifically instead of a random $c$.

## 5 EXPERIMENTS

We evaluate our method on the Phi-3-mini-4k-instruct (Abdin et al., 2024) and Meta-Llama-3.1-8B-Instruct (Dubey et al., 2024) models, and compare against GPTQ and greedy rounding (i.e. round-to-nearest, or RTN). We use the lm-evaluation-harness Gao et al. (2023) to evaluate on the Wikitext, GSM8k_cot 8-shot, MMLU 5-shot, ARC_Challenge 0-shot, PIQA 0-shot, HellaSwag 0-shot, and Winogrande 0-shot tasks. We report standard errors from lm-evaluation-harness. Wikitext measures perplexity, GSM8k is a generative task, and the remaining are multiple choice tasks. Note that generative tasks are typically more difficult than multiple choice tasks, and better reflect how the models are used in practice. See Appendix A for details on the hardware used, and hyper-parameter settings. Our method has similar memory requires as knowledge distillation, which also requires two copies of the model. We do not perform inference timing experiments; DiscQuant can optimize over a given quantization grid, so that we can utilize any pre-existing inference optimizations. For example, there are inference kernels for block scaling (Frantar et al., 2024) and incoherence processing (Tseng et al., 2024a). Ablations on the loss formulation are in Appendix A.

### 5.1 BLOCK SCALING

Our first experiments use standard block scaling quantization, determined by a `bits` and `groupsize` parameter. There are $2^{\texttt{bits}}$ unique points, and every `groupsize` parameters share a unique 16-bit scale parameter. For example, 3.25 bits is achieved with `bits=3, groupsize=64`. We use the block scaling implementation from Frantar et al. (2024) which is symmetric linear quantization. Table 2 shows the results quantizing Phi-3-mini-4k-instruct. Across all tasks and all bit settings, our method DiscQuant achieves superior or comparable compression over the baseline GPTQ and RTN methods. The gap between DiscQuant and the baselines is greater at lower bits. On the ARC_Challenge, PIQA, and WinoGrade tasks, DiscQuant achieves full recovery with at least 0.25 fewer bits per parameter than GPTQ and RTN. For example on ARC_Challenge, DiscQuant achieves full recovery at 4 bits per weight, whereas GPTQ requires 4.25 bits, and RTN 4.5 bits. DiscQuant achieves better compression on the more difficult generative GSM8k task: at 4 bits DiscQuant gets 77.3% accuracy, while GPTQ gets 71.5%, and RTN gets 62.2%. Table 3 shows the results quantizing Meta-Llama-3.1-8B-Instruct. Overall the story is the same. Our method DiscQuant achieves improved compression on the majority of quantization levels and tasks. For example at 4 bits, DiscQuant gets 66.5% GSM8k accuracy, while GPTQ gets 63.2%, and RTN gets 50.8%.

### 5.2 INCOHERENCE PROCESSING

We explore another quantization format to show that our method can compose with other quantization improvements. Incoherence processing has been shown to improve quantization, especially at less than 4 bits per weight Chee et al. (2023). The weights are multiplied

| Method | Wbits | Wiki↓ | GSM8k↑ | MMLU↑ | ArcC↑ | PIQA↑ | Hella↑ | Wino↑ |
|---|---|---|---|---|---|---|---|---|
| — | 16.0 | 9.5 | $84.4_{\pm1.0}$ | $70.4_{\pm0.4}$ | $56.7_{\pm1.4}$ | $80.8_{\pm0.9}$ | $77.4_{\pm0.4}$ | $73.5_{\pm1.2}$ |
| RTN | 3.0 | 6.3E5 | $1.0_{\pm0.3}$ | $23.3_{\pm0.4}$ | $26.9_{\pm1.3}$ | $53.4_{\pm1.2}$ | $28.2_{\pm0.4}$ | $48.6_{\pm1.4}$ |
| GPTQ | 3.0 | 28.2 | $2.3_{\pm0.4}$ | $37.7_{\pm0.4}$ | $34.8_{\pm1.4}$ | $64.3_{\pm1.1}$ | $56.5_{\pm0.5}$ | $52.6_{\pm1.4}$ |
| DiscQ | 3.0 | 17.7 | $26.8_{\pm1.2}$ | $45.6_{\pm0.4}$ | $44.1_{\pm1.5}$ | $73.9_{\pm1.0}$ | $63.3_{\pm0.5}$ | $66.6_{\pm1.3}$ |
| RTN | 3.25 | 22.5 | $31.0_{\pm1.3}$ | $53.2_{\pm0.4}$ | $48.4_{\pm1.5}$ | $72.5_{\pm1.0}$ | $68.3_{\pm0.5}$ | $62.6_{\pm1.4}$ |
| GPTQ | 3.25 | 13.8 | $54.3_{\pm1.4}$ | $59.0_{\pm0.4}$ | $49.6_{\pm1.5}$ | $77.3_{\pm1.0}$ | $71.1_{\pm0.5}$ | $66.5_{\pm1.3}$ |
| DiscQ | 3.25 | 12.6 | $64.2_{\pm1.3}$ | $60.7_{\pm0.4}$ | $53.5_{\pm1.5}$ | $78.7_{\pm1.0}$ | $72.3_{\pm0.4}$ | $72.5_{\pm1.3}$ |
| RTN | 3.5 | 18.8 | $46.3_{\pm1.4}$ | $57.0_{\pm0.4}$ | $46.2_{\pm1.5}$ | $73.8_{\pm1.0}$ | $70.0_{\pm0.5}$ | $63.9_{\pm1.4}$ |
| GPTQ | 3.5 | 12.8 | $54.6_{\pm1.4}$ | $61.7_{\pm0.4}$ | $51.6_{\pm1.5}$ | $78.9_{\pm1.0}$ | $72.3_{\pm0.4}$ | $68.3_{\pm1.3}$ |
| DiscQ | 3.5 | 12.0 | $69.5_{\pm1.3}$ | $63.0_{\pm0.4}$ | $51.1_{\pm1.5}$ | $78.9_{\pm1.0}$ | $73.0_{\pm0.4}$ | $73.9_{\pm1.2}$ |
| RTN | 4.0 | 14.6 | $62.2_{\pm1.3}$ | $61.2_{\pm0.4}$ | $53.6_{\pm1.5}$ | $76.3_{\pm1.0}$ | $72.9_{\pm0.4}$ | $65.3_{\pm1.3}$ |
| GPTQ | 4.0 | 11.5 | $71.5_{\pm1.2}$ | $65.1_{\pm0.4}$ | $54.6_{\pm1.5}$ | $78.8_{\pm1.0}$ | $74.7_{\pm0.4}$ | $70.9_{\pm1.3}$ |
| DiscQ | 4.0 | 11.2 | $77.3_{\pm1.2}$ | $65.7_{\pm0.4}$ | $56.8_{\pm1.4}$ | $79.5_{\pm0.9}$ | $74.5_{\pm0.4}$ | $72.0_{\pm1.3}$ |
| RTN | 4.25 | 11.2 | $64.4_{\pm1.3}$ | $67.5_{\pm0.4}$ | $55.5_{\pm1.5}$ | $79.3_{\pm0.9}$ | $76.1_{\pm0.4}$ | $69.1_{\pm1.3}$ |
| GPTQ | 4.25 | 10.3 | $81.0_{\pm1.1}$ | $68.5_{\pm0.4}$ | $56.9_{\pm1.4}$ | $79.7_{\pm0.9}$ | $76.1_{\pm0.4}$ | $72.1_{\pm1.3}$ |
| DiscQ | 4.25 | 10.2 | $80.7_{\pm1.1}$ | $68.4_{\pm0.4}$ | $57.3_{\pm1.4}$ | $80.7_{\pm0.9}$ | $76.3_{\pm0.4}$ | $74.2_{\pm1.2}$ |
| RTN | 4.5 | 10.8 | $71.6_{\pm1.2}$ | $67.7_{\pm0.4}$ | $57.5_{\pm1.4}$ | $79.3_{\pm0.9}$ | $76.6_{\pm0.4}$ | $72.2_{\pm1.3}$ |
| GPTQ | 4.5 | 10.1 | $82.0_{\pm1.1}$ | $68.8_{\pm0.4}$ | $55.8_{\pm1.5}$ | $80.8_{\pm0.9}$ | $76.5_{\pm0.4}$ | $71.8_{\pm1.3}$ |
| DiscQ | 4.5 | 10.0 | $82.1_{\pm1.1}$ | $68.5_{\pm0.4}$ | $56.6_{\pm1.4}$ | $80.2_{\pm0.9}$ | $76.7_{\pm0.4}$ | $74.2_{\pm1.2}$ |

Table 2: Phi-3-mini-4k-instruct. Across all tasks and bits, our method DiscQuant always achieves superior results over the baseline RTN and GPTQ methods. On the ArcC, PIQA, and Wino tasks, DiscQuant achieves full recovery with at least 0.25 fewer bits per parameter than GPTQ and RTN.

| Method | Wbits | Wiki↓ | GSM8k↑ | MMLU↑ | ArcC↑ | PIQA↑ | Hella↑ | Wino↑ |
|---|---|---|---|---|---|---|---|---|
| — | 16.0 | 8.7 | $77.0_{\pm1.2}$ | $68.0_{\pm0.4}$ | $55.2_{\pm1.5}$ | $81.3_{\pm0.9}$ | $79.3_{\pm0.4}$ | $73.7_{\pm1.2}$ |
| RTN | 3.0 | 4.4E3 | $0.5_{\pm0.2}$ | $23.2_{\pm0.4}$ | $22.3_{\pm1.2}$ | $52.4_{\pm1.2}$ | $29.1_{\pm0.5}$ | $50.0_{\pm1.4}$ |
| GPTQ | 3.0 | 23.2 | $3.6_{\pm0.5}$ | $24.6_{\pm0.4}$ | $31.8_{\pm1.4}$ | $66.6_{\pm1.1}$ | $45.8_{\pm0.5}$ | $54.1_{\pm1.4}$ |
| DiscQ | 3.0 | 15.2 | $14.3_{\pm1.0}$ | $44.6_{\pm0.4}$ | $39.4_{\pm1.4}$ | $73.2_{\pm1.0}$ | $64.4_{\pm0.5}$ | $62.8_{\pm1.4}$ |
| RTN | 3.25 | 15.2 | $10.8_{\pm0.9}$ | $50.5_{\pm0.4}$ | $44.3_{\pm1.5}$ | $75.2_{\pm1.0}$ | $71.4_{\pm0.5}$ | $67.2_{\pm1.3}$ |
| GPTQ | 3.25 | 10.7 | $56.3_{\pm1.4}$ | $60.5_{\pm0.4}$ | $46.3_{\pm1.5}$ | $76.7_{\pm1.0}$ | $74.4_{\pm0.4}$ | $68.7_{\pm1.3}$ |
| DiscQ | 3.25 | 10.5 | $58.3_{\pm1.4}$ | $60.2_{\pm0.4}$ | $49.1_{\pm1.5}$ | $79.1_{\pm0.9}$ | $75.1_{\pm0.4}$ | $72.1_{\pm1.3}$ |
| RTN | 3.5 | 12.7 | $35.9_{\pm1.3}$ | $51.4_{\pm0.4}$ | $48.4_{\pm1.5}$ | $76.7_{\pm1.0}$ | $73.0_{\pm0.4}$ | $69.1_{\pm1.3}$ |
| GPTQ | 3.5 | 10.4 | $57.0_{\pm1.4}$ | $62.1_{\pm0.4}$ | $49.9_{\pm1.5}$ | $77.3_{\pm1.0}$ | $75.1_{\pm0.4}$ | $71.1_{\pm1.3}$ |
| DiscQ | 3.5 | 10.3 | $60.7_{\pm1.3}$ | $60.9_{\pm0.4}$ | $51.7_{\pm1.5}$ | $79.2_{\pm0.9}$ | $76.3_{\pm0.4}$ | $72.5_{\pm1.3}$ |
| RTN | 4.0 | 12.5 | $50.8_{\pm1.4}$ | $59.3_{\pm0.4}$ | $50.5_{\pm1.5}$ | $77.6_{\pm1.0}$ | $74.7_{\pm0.4}$ | $69.9_{\pm1.3}$ |
| GPTQ | 4.0 | 9.9 | $63.2_{\pm1.3}$ | $64.4_{\pm0.4}$ | $52.4_{\pm1.5}$ | $78.4_{\pm1.0}$ | $75.9_{\pm0.4}$ | $71.7_{\pm1.3}$ |
| DiscQ | 4.0 | 9.8 | $66.5_{\pm1.3}$ | $63.4_{\pm0.4}$ | $51.6_{\pm1.5}$ | $79.2_{\pm0.9}$ | $76.9_{\pm0.4}$ | $72.8_{\pm1.3}$ |
| RTN | 4.25 | 9.4 | $70.6_{\pm1.3}$ | $65.7_{\pm0.4}$ | $54.2_{\pm1.5}$ | $80.1_{\pm0.9}$ | $78.0_{\pm0.4}$ | $73.9_{\pm1.2}$ |
| GPTQ | 4.25 | 9.1 | $74.6_{\pm1.2}$ | $66.8_{\pm0.4}$ | $53.4_{\pm1.5}$ | $79.6_{\pm0.9}$ | $77.9_{\pm0.4}$ | $73.5_{\pm1.2}$ |
| DiscQ | 4.25 | 9.1 | $74.9_{\pm1.2}$ | $66.9_{\pm0.4}$ | $53.6_{\pm1.5}$ | $79.9_{\pm0.9}$ | $78.4_{\pm0.4}$ | $72.6_{\pm1.3}$ |
| RTN | 4.5 | 9.3 | $71.9_{\pm1.2}$ | $65.8_{\pm0.4}$ | $54.8_{\pm1.5}$ | $80.3_{\pm0.9}$ | $78.4_{\pm0.4}$ | $72.4_{\pm1.3}$ |
| GPTQ | 4.5 | 9.0 | $73.8_{\pm1.2}$ | $66.9_{\pm0.4}$ | $53.6_{\pm1.5}$ | $79.6_{\pm0.9}$ | $78.1_{\pm0.4}$ | $73.7_{\pm1.2}$ |
| DiscQ | 4.5 | 9.1 | $74.8_{\pm1.2}$ | $66.8_{\pm0.4}$ | $54.1_{\pm1.5}$ | $80.6_{\pm0.9}$ | $78.7_{\pm0.4}$ | $72.9_{\pm1.2}$ |

Table 3: Meta-Llama-3.1-8B-Instruct. Our method DiscQuant achieves superior compression on the vast majority of quantization levels and tasks over the baselines GPTQ and RTN.

by certain random orthogonal matrices prior to quantization, which can reduce the range of the weights and make quantization easier. We employ the Randomized Hadamard Transform from Tseng et al. (2024a). We use the same block scaling quantization grid as in the previous subsection. A subset of our results are shown in Figure 5, where we superimpose bar

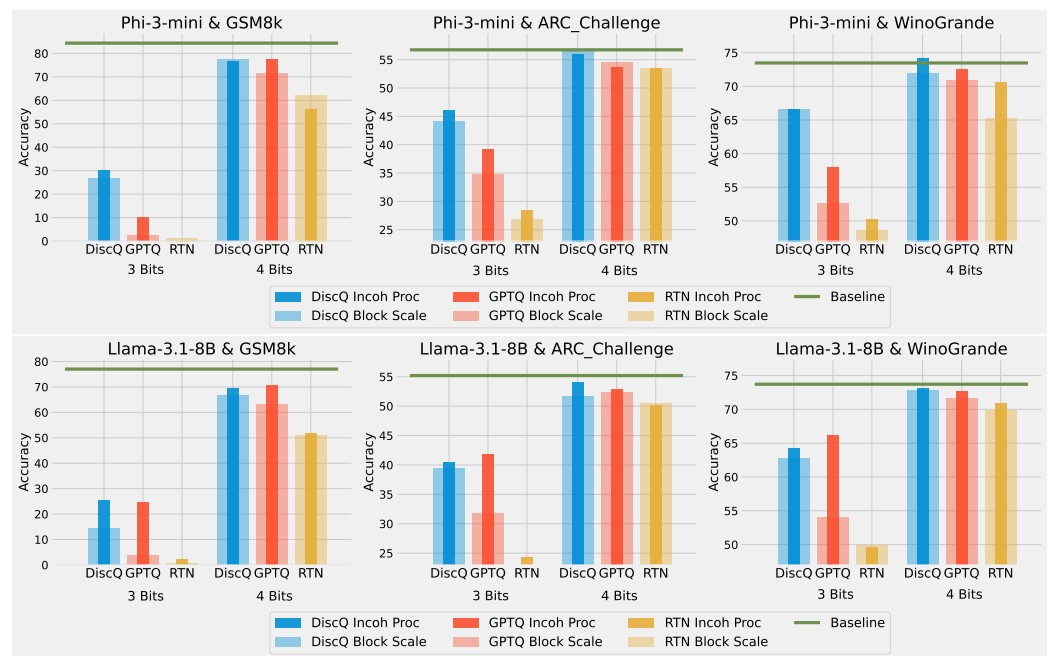

Figure 5: Quantizing Phi-3-mini-4k-instruct and Meta-LLama-3.1-8B-Instruct with block scaling, and additional incoherence processing. DiscQuant can compose with other quantization improvements, and with incoherence processing remains competitive with GPTQ.

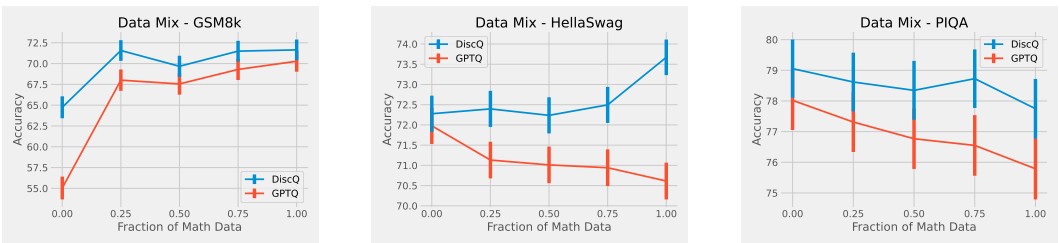

Figure 6: Effect of increasing the fraction of math data when quantizing Phi-3-mini-4k-instruct at 3.25 bits. For 8192 total samples, we use a fraction of math subject data (GSM8k & MetaMathQA), and the remaining our standard RedPajama. As expected, performance on GSM8k increases with more math data. Expected behavior on the other tasks is unclear.

plots for block scaling and block scaling + incoherence processing. In the majority of cases, adding incoherence processing increases the task accuracy, especially at lower bits. We do not use fractional bits, (i.e. no `groupsize`), due to the fact that both these methods effect outliers and can interfere with one another. Incoherence especially helps GPTQ at 3 bits, and for Phi-3 DiscQuant without incoherence is competitive to GPTQ with incoherence. For full results see Appendix A.

## 5.3 EFFECT OF DATA

We perform a simple investigation into the effect of the dataset on quantization. We mix math subeject data–GSM8k and MetaMathQA–with our standard RedPajama dataset. Figure 6 shows the results of quantizing Phi-3-mini-4k-instruct at 3.25 bits with such a mix. As expected, both methods increase accuracy on GSM8k when there is a greater fraction of math data. On HellaSwag, DiscQuant improves with more math data, where GPTQ gets worse. On PIQA, both methods get worse. See Appendix A for all tasks. There is a meaningful change in accuracy as a result of changing the data mix. Choosing an appropriate data mix for quantization remains an important open question.

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

| Method | Wbits | Wiki↓ | GSM8k↑ | MMLU↑ | ArcC↑ | PIQA↑ | Hella↑ | Wino↑ |
|--------|-------|-------|--------|-------|-------|-------|--------|-------|
| — | 16.0 | 9.5 | $84.4_{\pm1.0}$ | $70.4_{\pm0.4}$ | $56.7_{\pm1.4}$ | $80.8_{\pm0.9}$ | $77.4_{\pm0.4}$ | $73.5_{\pm1.2}$ |
| RTN | 3.0 | 2.6E5 | $0.0_{\pm0.0}$ | $23.4_{\pm0.4}$ | $28.5_{\pm1.3}$ | $49.8_{\pm1.2}$ | $26.0_{\pm0.4}$ | $50.2_{\pm1.4}$ |
| GPTQ | 3.0 | 20.8 | $10.0_{\pm0.8}$ | $43.8_{\pm0.4}$ | $39.2_{\pm1.4}$ | $70.5_{\pm1.1}$ | $60.7_{\pm0.5}$ | $58.0_{\pm1.4}$ |
| DiscQ | 3.0 | 16.7 | $29.9_{\pm1.3}$ | $48.0_{\pm0.4}$ | $46.2_{\pm1.5}$ | $75.1_{\pm1.0}$ | $64.5_{\pm0.5}$ | $66.6_{\pm1.3}$ |
| RTN | 4.0 | 15.9 | $56.3_{\pm1.4}$ | $55.0_{\pm0.4}$ | $53.5_{\pm1.5}$ | $77.4_{\pm1.0}$ | $68.8_{\pm0.5}$ | $70.6_{\pm1.3}$ |
| GPTQ | 4.0 | 11.0 | $77.6_{\pm1.1}$ | $65.8_{\pm0.4}$ | $53.7_{\pm1.5}$ | $80.2_{\pm0.9}$ | $74.9_{\pm0.4}$ | $72.5_{\pm1.3}$ |
| DiscQ | 4.0 | 11.0 | $76.7_{\pm1.2}$ | $65.6_{\pm0.4}$ | $56.0_{\pm1.5}$ | $79.5_{\pm0.9}$ | $74.9_{\pm0.4}$ | $74.2_{\pm1.2}$ |

Table 4: Phi-3-mini-4k-instruct with incoherence processing. At 3 bits per weight, DiscQuant achieves superior compression across all tasks. At 4 bits per weight, DiscQuant achieves comparable compression.

| Method | Wbits | Wiki↓ | GSM8k↑ | MMLU↑ | ArcC↑ | PIQA↑ | Hella↑ | Wino↑ |
|--------|-------|-------|--------|-------|-------|-------|--------|-------|
| — | 16.0 | 8.7 | $77.0_{\pm1.2}$ | $68.0_{\pm0.4}$ | $55.2_{\pm1.5}$ | $81.3_{\pm0.9}$ | $79.3_{\pm0.4}$ | $73.7_{\pm1.2}$ |
| RTN | 3.0 | 2.4E3 | $2.1_{\pm0.4}$ | $25.2_{\pm0.4}$ | $24.3_{\pm1.3}$ | $54.7_{\pm1.2}$ | $29.5_{\pm0.5}$ | $49.6_{\pm1.4}$ |
| GPTQ | 3.0 | 13.9 | $24.4_{\pm1.2}$ | $49.7_{\pm0.4}$ | $41.7_{\pm1.4}$ | $73.1_{\pm1.0}$ | $70.4_{\pm0.5}$ | $66.2_{\pm1.3}$ |
| DiscQ | 3.0 | 13.4 | $25.4_{\pm1.2}$ | $51.5_{\pm0.4}$ | $40.4_{\pm1.4}$ | $73.2_{\pm1.0}$ | $69.6_{\pm0.5}$ | $64.2_{\pm1.3}$ |
| RTN | 4.0 | 11.2 | $51.6_{\pm1.4}$ | $59.5_{\pm0.4}$ | $50.1_{\pm1.5}$ | $78.9_{\pm1.0}$ | $74.5_{\pm0.4}$ | $71.0_{\pm1.3}$ |
| GPTQ | 4.0 | 9.5 | $70.7_{\pm1.3}$ | $64.9_{\pm0.4}$ | $52.8_{\pm1.5}$ | $80.0_{\pm0.9}$ | $77.4_{\pm0.4}$ | $72.7_{\pm1.3}$ |
| DiscQ | 4.0 | 9.6 | $69.4_{\pm1.3}$ | $63.7_{\pm0.4}$ | $54.1_{\pm1.5}$ | $80.7_{\pm0.9}$ | $77.0_{\pm0.4}$ | $73.2_{\pm1.2}$ |

Table 5: Meta-Llama-3.1-8B-Instruct with incoherence processing. Across a majority of bits and tasks, DiscQuant achieves comparable compression with GPTQ, and does better than RNT.

## A ADDITIONAL EXPERIMENTS

### A.1 EXPERIMENTAL SETUP DETAILS

The experiments for the Phi-3-mini model were conducted on either a single 80GB Nvidia A100 GPU, or 2x40GB A100 GPUs, while the Llama-3.1-8B model used either 2x80GB A100s, or 4x40GB A100s. We use the PyTorch framework. We initialize $x \in [0,1]^n$ uniformly at random, and used AdamW (Loshchilov & Hutter, 2019) with a cosine learning rate schedule. We multiply the regularization coefficient $\lambda$ with the KL loss term, and perform entry-wise gradient clipping on the KL loss term. For DiscQuant, we tuned the hyper-parameters for each model and bit setting. The hyper-parameters `clamp`, $\lambda$, `lr`, `batch_size`, `num_iter` and `warmup` were tuned. In the block scaling setting we found that `clamp={1.0, 0.5}`, $\lambda$=200, `lr={0.1, 0.05}`, `batch_size={4,8}`, `num_iter`=1024, `warmup`=128 worked well for both models. In the incoherence processing setting we found that `clamp={0.05,0.01}`, `lr={0.05,0.01}` worked well for both models, all other parameters being the same as before. For GPTQ, we used the `actorder`, `true_sequential` heuristics, and as tuned the number of samples over {1024, 4096, 8192} for each model and bit setting. Our quantization dataset is constructed from the RedPajama-1T-Sample training set (Computer, 2023). We concatenate random samples until up to 2048 sequence length, truncating the last sample if necessary. Greedy or round-to-nearest requires no data, and no hyper-parameter tuning.

### A.2 INCOHERENCE PROCESSING

Table 4 shows our results quantizing Phi-3-mini-4k-instruct with incoherence processing. At 3 bits per weight, DiscQuant achieves superior compression across all tasks. At 4 bits per weight, DiscQuant achieves comparable compression. For example, on ARC_CHallenge at 3 bits, DiscQuant achieves 46.2% accuracy, while GPTQ achieves 39.2%, and RTN 28.5%. Table 5 shows our results quantizing Meta-Llama-3.1-8B-Instruct with incoherence processing. DiscQuant performs comparably to GPTQ, and better than RTN. For example, on

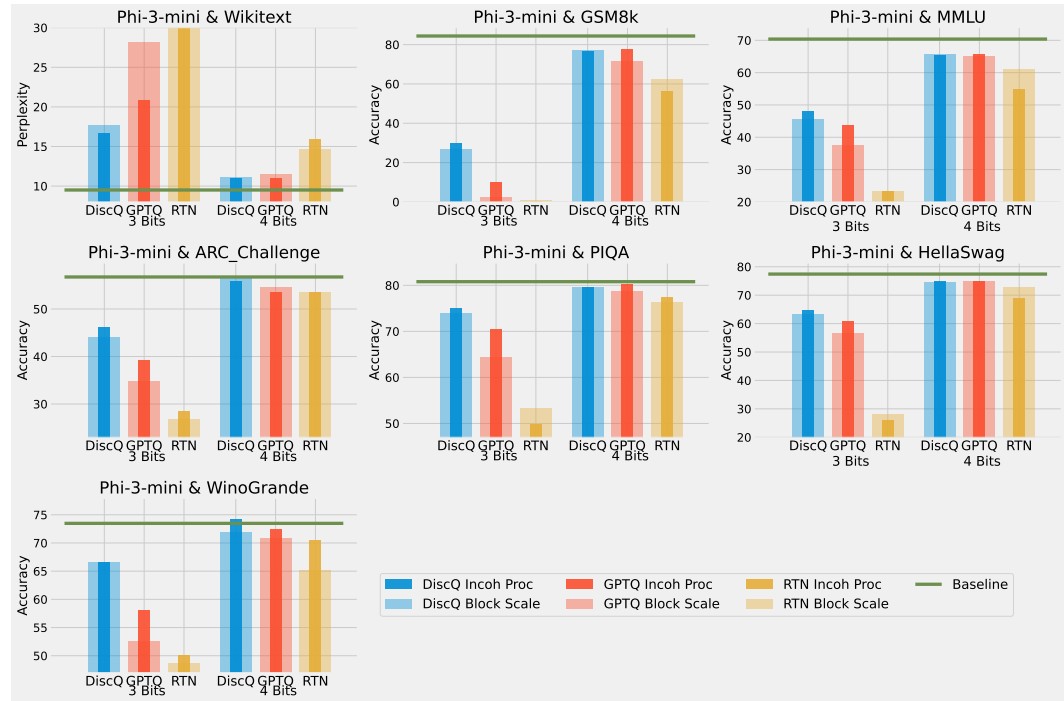

Figure 7: Quantizing Phi-3-mini-4k-instruct with block scaling, and additional incoherence processing. Adding incoherence processing largely improves model quality at 3 bits. At 4 bits, these improvements are smaller. At 3 bits, DiscQuant is better than GPTQ with incoherence processing.

WinoGrande at 4 bits, DiscQuant achieves 73.2% accuracy, while GPTQ achieves 72.7%, and RTN 71.0%.

Figures 7 and 8 show the results adding incoherence processing superimposed over just using block scaling. Incoherence processing largely improves quantization at 3 bits across both models, whereas at 4 bits the improvements are smaller. In the Phi-3 model at 3 bits, DiscQuant without incoherence is better than GPTQ with incoherence. Across the other models and bit settings, DiscQuant and GPTQ are comparable after incoherence processing.

## A.3    Effect of Data

Here we give the full set of evaluation tasks when changing the mix of math subject data when quantizing Phi-3-mini-4k-instruct to 3.25 bits. It is interesting that across all evaluation tasks, there is a meaningful change in evaluation metrics as a result of changing the data mix. We leave the question of appropriate data curation as an important open question.

## A.4    Ablations

We tried several distillation formulations, but ultimately chose a standard KL divergence between the outputs of the original and quantized model as the best approach. See Table 6. We quantize Phi-3-mini-4k-instruct to 3.25 bits, using 1024 samples. We tune the hyperparameters as described at the beginning of this section. Note that for these ablations we used fewer samples than in our main experiments. In addition to the standard KL divergence, we tried several intermediate loss formulations for knowledge distillation. We used a normalized L2 loss between the outputs of the teacher and student, either per decoder layer (Intermed Type = Layer), or between each linear layer (Intermed Type = Linear). This distillation formulation was presented in Kurtic et al. (2023) for recovering LLMs after pruning. We also investigated taking an affine combination between the KL and intermediate

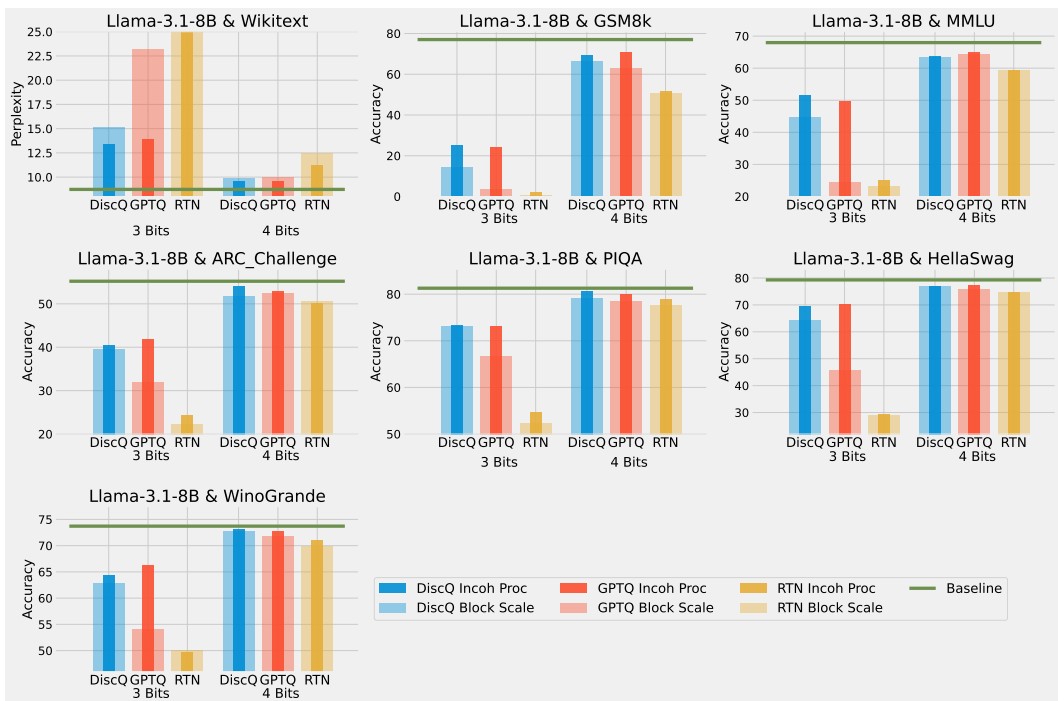

Figure 8: Quantizing Meta-Llama-3.1-8B-Instruct with block scaling, and additional incoherence processing. Adding incoherence processing largely improves model quality at 3 bits. At 4 bits, these improvements are smaller. After incoherence, DiscQuant is largely comparable to GPTQ.

losses, trying several different coefficients. Table 6 shows our results; using just the KL divergence gives the best results. We also tried minimizing the ground truth loss instead of a distillation loss. We use the same setup as Table 6, and find that minimizing the ground truth loss achieves 52.7% GSM8k accuracy, and 13.6 Wikitext perplexity. Therefore we use the KL divergence.

# B ROUNDING WEIGHTS VIA DISCREPANCY THEORY

## B.1 THE LOVETT MEKA ALGORITHM

A seminal result by Lovett and Meka Lovett & Meka (2012) works as follows: we are given a point $y \in [0,1]^n$ in the hypercube, vectors $v_1, \ldots, v_m \in \mathbb{R}^n$ with $\|v_j\|_2 = 1$ and parameters $c_j \geq 0$ so that $\sum_{j=1}^m e^{-c_j^2/16} \leq \frac{n}{16}$. Then in randomized polynomial time one can find a point $x \in [0,1]^n$ so that $|\langle v_j, x - y \rangle| \leq c_j$ for all $j$ and at least half the coordinates of $x$ are integral. Their algorithm is simple and elegant: we construct $x$ as the outcome of a random walk starting at $y$. Then iteratively, for some small step size $\delta > 0$ we add the outcome of a random Gaussian times $\delta$ to the current point. After hitting some constraint $x_i = 0$, $x_i = 1$ or $|\langle v_j, x - y \rangle| = c_j$, the Gaussian updates will be taken orthogonal to those normal vectors. In other words, the random walk will continue in the face of the described polytope. Still Lovett & Meka (2012) prove that performing the updates for $O(\frac{1}{\delta^2})$ iterations the walk will cover enough distance so that on average $\Theta(n)$ box constraints must become tight.

In our setting we only need to use parameters $c_j = 0$. However we use some properties of the Lovett-Meka algorithm that are not explicitly stated elsewhere. Here we denote $\|M\|_{\mathcal{S}(1)}$ as the sum of the singular values of a matrix $M$ (also called *Schatten-1 norm*, *nuclear norm* or *trace norm* of $M$).

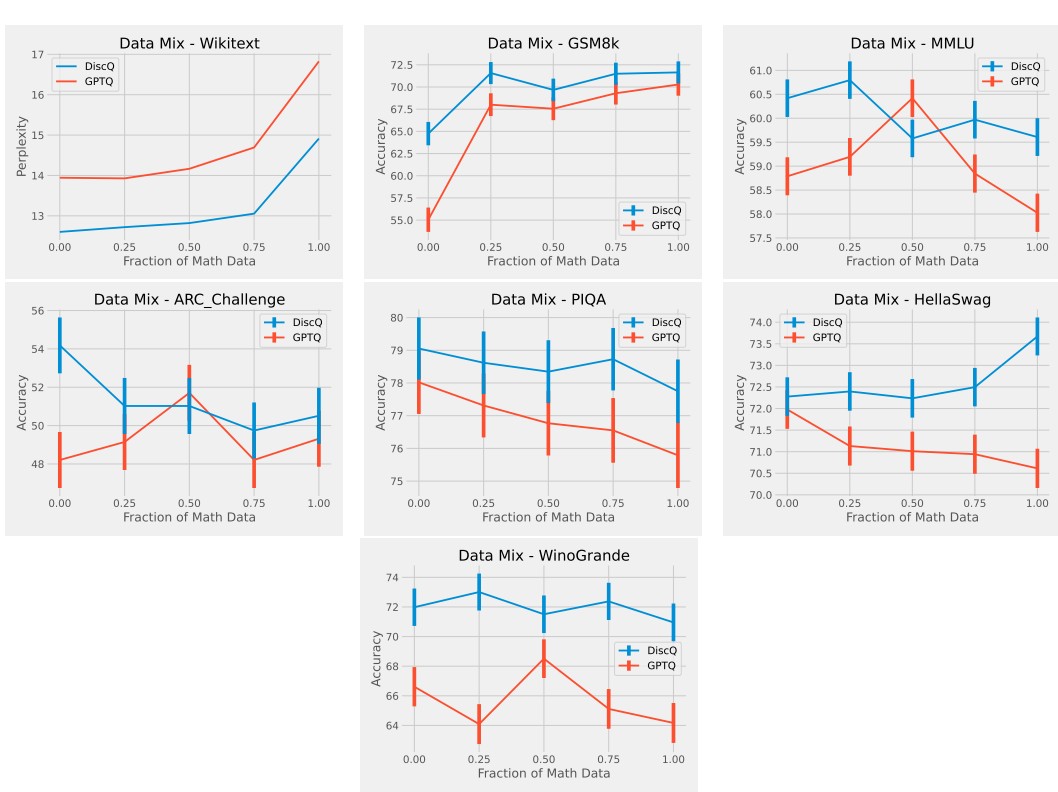

Figure 9: Effect of increasing the fraction of math data when quantizing Phi-3-mini-4k-instruct at 3.25 bits. For 8192 total samples, we use a fraction of math subject data (GSM8k and MetaMathQA), and the remaining our standard RedPajama. Across all evaluations, there is a meaningful change as a result of changing the data mix.

| KL Coeff | Intermed Coeff | Intermed Type | Wiki↓ | GSM8k↑ |
|----------|----------------|---------------|-------|--------|
| 1.0 | 0.0 | None | 12.8 | $64.9_{\pm 1.3}$ |
| 0.0 | 1.0 | Layer | 14.7 | $54.1_{\pm 1.4}$ |
| 0.0 | 1.0 | Linear | 14.3 | $60.1_{\pm 1.4}$ |
| 0.1 | 0.9 | Linear | 13.1 | $61.4_{\pm 1.3}$ |
| 0.5 | 0.5 | Linear | 12.9 | $63.9_{\pm 1.3}$ |
| 0.9 | 0.1 | Linear | 12.8 | $63.8_{\pm 1.3}$ |

Table 6: Distillation Ablations. Quantizing Phi-3-mini-4k to 3.25 bits using a reduced 1024 samples of RedPajama. We test affine combinations between the KL divergence loss and intermediate L2 loss, which is either between the linear or decoder layers. Standard KL divergence does best.

**Theorem B.1** (Derived from Lovett & Meka (2012)). *Let $g_1, \ldots, g_m \in \mathbb{R}^n$ be any vectors with $m \leq \frac{n}{16}$ and let $y \in [0,1]^n$. Then in polynomial time one can compute a sample $x \sim \mathcal{D} := \mathcal{D}_{LM}(g_1, \ldots, g_m, y)$ so that*

*(i) One has $x \in [0,1]^n$ and with probability at least $\frac{1}{10}$ one has $|\{j \in [n] : x_j \in \{0,1\}\}| \geq \frac{n}{2}$.*

*(ii) For any vector $\theta \in \mathbb{R}^n$ one has $\mathbb{E}_{x \sim \mathcal{D}}[\langle \theta, x - y \rangle^2] \leq O(\|\theta\|_2^2)$.*

*(iii) For any symmetric matrix $M \in \mathbb{R}^{n \times n}$ one has $\mathbb{E}[\langle M, (x-y)(x-y)^T \rangle] \leq O(\|M\|_{\mathcal{S}(1)})$.*

*Proof.* (i) is explicitly in Lovett & Meka (2012). For (ii) we use that the outcome of the random walk is of the form

$$x = y + \delta \sum_{t=1}^{O(1/\delta^2)} u_t \quad \text{where} \quad u_t \sim N(\mathbf{0}, \Sigma_t)$$

Here $0 \preceq \Sigma_t \preceq I_n$. But crucially each covariance matrix $\Sigma_t$ may depend on the outcome of $u_1, \ldots, u_{t-1}$. In particular it is *not* true that $x - y$ is Gaussian. But it is a Martingale and as for each step $t$ one has $\mathbb{E}[\langle u_t, \theta \rangle] = 0$ and $\mathbb{E}[\langle u_t, \theta \rangle^2] \leq O(\|\theta\|_2^2)$, the variance still satisfies $\mathbb{E}[\langle \delta \sum_{t=1}^{O(1/\delta^2)} u_t, \theta \rangle^2] \leq O(\|\theta\|_2^2)$ which settles (ii). Finally we argue why (iii) holds. We note that (ii) can be restated as $\mathbb{E}_{x \sim \mathcal{D}}[(x-y)(x-y)^T] \preceq O(1) \cdot I_n$. Then

$$\mathbb{E}[\langle M, (x-y)(x-y)^T \rangle] = \langle M, \mathbb{E}[(x-y)(x-y)^T] \rangle$$
$$\leq \|M\|_{\mathcal{S}(1)} \cdot \|\mathbb{E}[(x-y)(x-y)^T]\|_{\text{op}}$$
$$\leq O(\|M\|_{\mathcal{S}(1)}).$$

$\square$

### B.2 The main theoretical result

As explained earlier we assume that we are given a weight vector $y \in [0,1]^n$ and have access to samples $g_1, \ldots, g_m \sim \mathcal{D}$ where $\mathcal{D}$ is a distribution on $\mathbb{R}^n$ whose covariance matrix $\Sigma := \mathbb{E}_{g \sim \mathcal{D}_{\text{data}}}[gg^T]$ has rapidly decaying Eigenvalues, say $\lambda_k \leq \frac{C}{k^\alpha}$ for some constants $C > 0$ and $\alpha > 1$. In order to prove rigorous bounds we also need a mild assumption that provides that the distribution is well-behaved. We use the notion by O'Donnell O'Donnell (2014) and say that for a parameter $\beta \geq 1$, a random vector $X \in \mathbb{R}^n$ is *$\beta$-reasonable* if

$$\mathbb{E}[\langle X, \theta \rangle^4] \leq \beta \cdot \mathbb{E}[\langle X, \theta \rangle^2]^2 \quad \forall \theta \in \mathbb{R}^n$$

For example $X \sim \{-1, 1\}^n$ and a Gaussian $X \sim N(\mathbf{0}, \Sigma)$ are both $O(1)$-reasonable. Our main theoretical result is then:

**Theorem B.2.** *Let $\alpha > 1$ and $\beta \geq 1$ be constants and let $1 \leq m \leq \frac{n}{16}$. Let $\mathcal{D}$ be a $\beta$-reasonable distribution with unknown covariance matrix $\Sigma \in \mathbb{R}^{n \times n}$ whose Eigenvalues satisfy $\lambda_k \leq \frac{1}{k^\alpha}$ for all $k = 1, \ldots, n$. Then there is a randomized polynomial time algorithm that given a $y \in [0,1]^n$ and $m$ independent samples $g_1, \ldots, g_m \sim \mathcal{D}$, produces an $x \in [0,1]^n$ so that with probability at least 0.99 one has*

*(i) $|frac(x)| \leq 16m$*

*(ii) $\langle \Sigma, (x-y)(x-y)^T \rangle \lesssim_\alpha \log(\frac{n}{m}) \cdot F_\alpha(m,n)$ where*

$$F_\alpha(m,n) := \begin{cases} m^{1-\alpha} & \text{if } 1 < \alpha < \frac{3}{2} \\ \frac{\log(n)}{\sqrt{m}} & \text{if } \alpha = \frac{3}{2} \\ \frac{1}{\sqrt{m}} & \text{if } \alpha > 3/2. \end{cases}$$

Ignoring polylogarithmic factors, this means that we can find an $x$ with $O(m)$ fractional coordinates left and $\langle \Sigma, (x-y)(x-y)^T \rangle \leq \max\{m^{1-\alpha}, \frac{1}{\sqrt{m}}\}$. The algorithm to compute $x$ as in Theorem B.2 is simple:

---

LOVETT-MEKA ROUNDING ALGORITHM

---

**Input:** Weight vector $y \in [0,1]^n$ and parameter $m$
**Output:** Rounded vector $x$

   (1) Sample $g_1, \ldots, g_m \sim \mathcal{D}$. Initialize $x^{(0)} := y$

   (2) FOR $t = 1$ TO $\infty$ DO

       (3) IF $|\text{frac}(x^{(t-1)})| \leq 16m$ then return $x^{(t-1)}$
       (4) Set $x^{(t)} := \mathcal{D}_{LM}(g_1, \ldots, g_m, x^{(t-1)})$

---

A crucial aspect of analyzing this algorithm is understanding how far the *covariance estimator* $\frac{1}{m}\sum_{j=1}^m g_j g_j^T$ is from the actual covariance matrix $\Sigma$ in terms of the *Schatten 1-norm* $\|\cdot\|_{\mathcal{S}(1)}$. We use the following result.

**Proposition B.3.** *Let $\alpha > 1$, $\beta \geq 1$ and let $\mathcal{D}$ be a $\beta$-reasonable distribution with covariance matrix $\Sigma \in \mathbb{R}^{n \times n}$ whose Eigenvalues satisfy $\lambda_k \leq \frac{1}{k^\alpha}$ for all $k = 1, \ldots, n$. Let $g_1, \ldots, g_m \sim \mathcal{D}$ be independent samples and let $X^{(\ell)} := \frac{1}{m} g_\ell g_\ell^T$ and $X := \sum_{\ell=1}^m X^{(\ell)}$. Then*

$$\mathbb{E}[\|X - \Sigma\|_{\mathcal{S}(1)}] \lesssim_{\alpha,\beta} F_\alpha(m,n)$$

*where $F_\alpha(m,n)$ is as defined in Theorem B.2.*

We postpone the proof of Prop B.3 to Section B.3 and first conclude the proof of Theorem B.2.

*Proof of Theorem B.2.* Suppose $x^{(t^*)}$ is the vector that the algorithm returned in (3). It will be notationally convenient to define $x^{(t)} := x^{(t^*)}$ for all $t > t^*$. We say that iteration $t$ is *good* if either $|\text{frac}(x^{(t-1)})| \leq 16m$ or if $|\text{frac}(x^{(t)})| \leq \frac{1}{2}|\text{frac}(x^{(t-1)})|$. If an iteration $t$ is not good, we repeat the iteration until it is good. From Theorem B.1.(i) we know that every iteration is good with probability at least $\frac{1}{10}$ (independently of previous outcomes), thus by standard Chernov bounds, with probability at least 0.99, within the first $T := C' \log(\frac{n}{m})$ iterations there must be at least $\log(\frac{n}{m})$ many good iterations, for $C' > 0$ a sufficiently large constant. After $\log(\frac{n}{m})$ good iterations, one has $|\text{frac}(x^{(T)})| \leq 16m$, and moreover the suffered discrepancy is

$$\mathbb{E}\left[\langle \Sigma, (x^{(T)} - y)(x^{(T)} - y)^T \rangle\right] \leq \sum_{t=1}^T \mathbb{E}\left[\langle \Sigma, (x^{(t)} - x^{(t-1)})(x^{(t)} - x^{(t-1)})^T \rangle\right] \lesssim_{\alpha,\beta} T \cdot F_\alpha(m,n).$$

Thus the claim then follows. $\qquad\square$

### B.3 Analyzing the covariance estimator

It remains to prove Prop B.3.

*Proof of Prop B.3.* We first present the proof for the case of $1 < \alpha < \frac{3}{2}$ and then discuss the modifications for the other two cases. The claim is invariant under a change of basis, hence we may assume that $\Sigma$ is a diagonal matrix with Eigenvalues $\lambda_1 \geq \ldots \geq \lambda_n \geq 0$, i.e. $\Sigma_{ii} = \lambda_i$ for all $i \in [n]$. We can bound the variance terms for all entries (whether diagonal or not):

**Claim I.** *For all $i, j \in [n]$ one has $\mathbb{E}[|X_{ij} - \Sigma_{ij}|^2] \lesssim_\beta \frac{\lambda_i \lambda_j}{m}$.*

**Proof of Claim I.** We recall that $\mathbb{E}[X] = \Sigma$ and $\mathbb{E}[X^{(\ell)}] = \frac{1}{m}\Sigma$. For all $i, j \in [n]$ one has

$$
\begin{aligned}
\mathbb{E}[|X_{ij} - \Sigma_{ij}|^2] &= \text{Var}[X_{ij}] \\
&= \sum_{\ell=1}^{m} \text{Var}[X_{ij}^{(\ell)}] \\
&= \frac{1}{m}\mathbb{E}_{h \sim \mathcal{D}}[|h_i h_j - \Sigma_{ij}|^2] \\
&\leq \frac{2}{m}\Big(\mathbb{E}_{h \sim \mathcal{D}}[h_i^2 h_j^2] + \underbrace{\Sigma_{ij}^2}_{\leq \lambda_i \lambda_j}\Big) \\
&\overset{(*)}{\leq} \frac{2}{m}(\mathbb{E}_{h \sim \mathcal{D}}[h_i^4]^{1/2}\mathbb{E}_{h \sim \mathcal{D}}[h_j^4]^{1/2} + \lambda_i \lambda_j) \\
&\overset{(**)}{\leq} \frac{2}{m}\big(\beta^{1/2}\underbrace{\mathbb{E}_{h \sim \mathcal{D}}[h_i^2]}_{=\lambda_i} \cdot \beta^{1/2}\underbrace{\mathbb{E}_{h \sim \mathcal{D}}[h_j^2]}_{=\lambda_j} + \lambda_i \lambda_j\big) = \frac{2\beta + 2}{m} \cdot \lambda_i \lambda_j
\end{aligned}
$$

Here we use the inequality $(a - b)^2 \leq 2a^2 + 2b^2$. Moveover $\Sigma_{ij} \leq \lambda_i \lambda_j$ holds because $\Sigma$ is a diagonal matrix. Note that we have used Cauchy-Schwarz in $(*)$ and the assumption that $\mathcal{D}$ is $\beta$-reasonable in $(**)$. $\qquad\square$

Now let $J_\ell := \{i \in [n] \mid 2^{\ell-1} \leq i < 2^\ell\}$. It will be useful to note that $|J_\ell| \leq 2^\ell$ and the sum of the Eigenvalues in each block satisfies $\sum_{i \in J_\ell} \lambda_i \lesssim 2^\ell \cdot (2^\ell)^{-\alpha} = (2^\ell)^{1-\alpha}$. Our strategy is to use the triangle inequality to bound:

$$
\mathbb{E}[\|X - \Sigma\|_{\mathcal{S}(1)}] \leq 2 \sum_{\ell \geq 1} \sum_{k \geq \ell} \mathbb{E}[\|X_{J_\ell, J_k} - \Sigma_{J_\ell, J_k}\|_{\mathcal{S}(1)}] \tag{4}
$$

Here $X_{J_\ell, J_k}$ is the $|J_\ell| \times |J_k|$ submatrix of $X$ that is indexed by rows $J_\ell$ and columns $J_k$. In the following we will estimate the contribution of the different blocks depending on their parameter regime and whether they are diagonal or off-diagonal.

**Claim II.** *Let $\ell \leq k$ and abbreviate $Y := X_{J_\ell, J_k} - \Sigma_{J_\ell, J_k}$. Then*

$$
\mathbb{E}[\|Y\|_{\mathcal{S}(1)}] \lesssim \sqrt{\frac{r}{m}} \cdot 2^{\frac{\ell+k}{2}(1-\alpha)}
$$

*assuming that $rank(Y) \leq r$ for any outcome of $Y$.*

**Proof of Claim II.** We recall that for any matrix $A$ one has $\|A\|_{\mathcal{S}(1)} \leq \sqrt{\text{rank}(A)} \cdot \|A\|_F$. Then for all $\ell \leq k$ we can bound

$$
\begin{aligned}
\mathbb{E}[\|Y\|_{\mathcal{S}(1)}] &\leq \sqrt{r} \cdot \mathbb{E}[\|Y\|_F] \\
&\overset{\text{Jensen}}{\leq} \sqrt{r} \cdot \mathbb{E}[\|Y\|_F^2]^{1/2} \\
&\overset{\text{Claim I}}{\lesssim_\beta} \sqrt{r} \cdot \Big(\frac{1}{m}\Big(\sum_{i \in J_\ell} \lambda_i\Big)\Big(\sum_{j \in J_k} \lambda_j\Big)\Big)^{1/2} \\
&\lesssim \sqrt{r} \cdot \sqrt{\frac{1}{m} \cdot (2^\ell)^{1-\alpha} \cdot (2^k)^{1-\alpha}} \\
&= \sqrt{\frac{r}{m}} \cdot 2^{\frac{\ell+k}{2}(1-\alpha)} \qquad\square
\end{aligned}
$$

Now we can bound the contribution that off-diagonal blocks have to Eq (4). Here we use that $\Sigma_{J_\ell, J_k} = \mathbf{0}$ and $\mathrm{rank}(X_{J_\ell, J_k}) \leq \min\{m, 2^\ell\}$. Then

$$
\sum_{\ell \geq 1} \sum_{k > \ell} \mathbb{E}\big[\|X_{J_\ell, J_k} - \underbrace{\Sigma_{J_\ell, J_k}}_{=\mathbf{0}}\|_{\mathcal{S}(1)}\big] \overset{\text{Claim II}}{\leq} \sum_{\ell \geq 1} \sum_{k > \ell} \frac{\sqrt{\min\{m, 2^\ell\}}}{\sqrt{m}} \cdot 2^{\frac{\ell+k}{2}(1-\alpha)}
$$

$$
= \sum_{\ell \geq 1} \min\big\{1, \sqrt{2^\ell/m}\big\} \cdot 2^{\frac{\ell}{2}(1-\alpha)} \underbrace{\sum_{k > \ell} \cdot 2^{\frac{k}{2}(1-\alpha)}}_{\lesssim_\alpha 2^{\ell(1-\alpha)/2}}
$$

$$
\lesssim_\alpha \sum_{\ell \geq 1} \min\big\{1, \sqrt{2^\ell/m}\big\} \cdot (2^\ell)^{1-\alpha} \tag{5}
$$

$$
\lesssim_\alpha m^{1-\alpha}
$$

In the last step we use that the function $z \mapsto \sqrt{z} \cdot z^{1-\alpha}$ is monotonically increasing while $z \mapsto z^{1-\alpha}$ is monotonically decreasing as we assume that $1 < \alpha < \frac{3}{2}$. Hence the term with $m = 2^\ell$ dominates the sum.

It remains to bound the diagonal blocks. First we consider the regime of small indices. Here we use the bound $\mathrm{rank}(X_{J_\ell, J_\ell} - \Sigma_{J_\ell, J_\ell}) \leq |J_\ell| \leq 2^\ell$ which gives

$$
\sum_{\ell : 2^\ell \leq m} \mathbb{E}[\|X_{J_\ell, J_\ell} - \Sigma_{J_\ell, J_\ell}\|_{\mathcal{S}(1)}] \overset{\text{Claim II}}{\leq} \sum_{\ell : 2^\ell \leq m} \sqrt{\frac{2^\ell}{m}} \cdot 2^{\ell(1-\alpha)} \lesssim m^{1-\alpha} \tag{6}
$$

Here the last summand (with $2^\ell = m$) dominates the sum in (6), again as $z \mapsto \sqrt{z} \cdot z^{1-\alpha}$ is monotonically increasing.

The final regime to consider is the one of large indices, i.e. diagonal blocks with $2^\ell > m$. In that case we can ignore any concentration that the randomness may provide and simply bound

$$
\sum_{\ell : 2^\ell > m} \mathbb{E}[\|X_{J_\ell, J_\ell} - \Sigma_{J_\ell, J_\ell}\|_{\mathcal{S}(1)}] \leq \sum_{\ell : 2^\ell > m} \big(\mathbb{E}[\|X_{J_\ell, J_\ell}\|_{\mathcal{S}(1)}] + \|\Sigma_{J_\ell, J_\ell}\|_{\mathcal{S}(1)}\big)
$$

$$
= \sum_{\ell : 2^\ell > m} \big(\mathbb{E}[\mathrm{Tr}[X_{J_\ell, J_\ell}]] + \mathrm{Tr}[\Sigma_{J_\ell, J_\ell}]\big)
$$

$$
= \sum_{j=m}^{n} \big(\underbrace{\mathbb{E}[X_{jj}]}_{=\Sigma_{jj}} + \underbrace{\Sigma_{jj}}_{\leq j^{-\alpha}}\big)
$$

$$
\lesssim \sum_{j \geq m} \frac{1}{j^\alpha} \lesssim m^{1-\alpha} \tag{7}
$$

Here we use again the triangle inequality of the trace norm and the fact that the matrices $X_{J_\ell, J_\ell}$ and $\Sigma_{J_\ell, J_\ell}$ are always positive semidefinite. This concludes the argument for $1 < \alpha < \frac{3}{2}$. If $\alpha = \frac{3}{2}$ then $\sqrt{2^\ell/m} \cdot (2^\ell)^{1-\alpha} \leq \frac{1}{\sqrt{m}}$ for each $\ell \geq 1$ and so (5) is bounded by $\frac{\log(n)}{\sqrt{m}}$. Moreover, the last two cases can be merged as

$$
\sum_{\ell \geq 1} \mathbb{E}[\|X_{J_\ell, J_\ell} - \Sigma_{J_\ell, J_\ell}\|_{\mathcal{S}(1)}] \overset{\text{Claim II}}{\leq} \sum_{\ell \geq 1} \sqrt{\frac{2^\ell}{m}} \cdot 2^{\ell(1-\alpha)} \lesssim \frac{\log(n)}{\sqrt{m}} \tag{8}
$$

Finally, if $\alpha > \frac{3}{2}$ then the first term (for $\ell = 1$) dominates the sums in (5) and (8) and the extra $\log(n)$ term can be omitted. $\qquad\square$

## C  Non-uniform Quantization Grid

We will introduce a new parameters $x \in [0, 1]^n$ and define

$$
w^x = w^{\text{down}} \odot (1 - x) + w^{\text{up}} \odot x
$$

where $\odot$ is component-wise product. Note that $w_i^x$ interpolates between $w_i^{\mathrm{down}}$ and $w_i^{\mathrm{up}}$ where $w_i = w_i^{\mathrm{down}}$ if $x_i = 0$ and $w_i = w_i^{\mathrm{up}}$ if $x_i = 1$. Let $y \in [0,1]^n$ be the interpolation point corresponding to the original weights, i.e., $w^y = w$. We can rewrite the linear constraints in terms of $x$ as follows:

$$\begin{aligned}
\langle \nabla_w f(w; s_i), w^x - w \rangle &= \langle \nabla_w f(w; s_i), w^x - w^y \rangle \\
&= \langle \nabla_w f(w; s_i), (w^{\mathrm{up}} - w^{\mathrm{down}}) \odot (x - y) \rangle \\
&= \langle \nabla_w f(w; s_i) \odot (w^{\mathrm{up}} - w^{\mathrm{down}}), x - y \rangle .
\end{aligned}$$

Let $M$ be an $m \times n$ matrix whose $i^{th}$ row is given by $\nabla_w f(w; s_i) \odot (w^{\mathrm{up}} - w^{\mathrm{down}})$. Then the linear constraints can be simply written as $M(x - y) = 0$.

## D  TAYLOR SERIES FOR KL DIVERGENCE

Let $p_w(\cdot|z_{<i})$ be the distribution of the next token predicted by the original model given prefix $z_{<i}$ where $z \sim \mathcal{D}_{\mathrm{data}}$ is a sample from the data distribution. Let

$$\mathrm{error}(\hat{w}) = \mathbb{E}_{z \sim \mathcal{D}_{\mathrm{data}}} \mathbb{E}_i D_{KL} \left( p_w(\cdot|z_{<i}) \,\|\, p_{w^x}(\cdot|z_{<i}) \right)$$

be the KL divergence between the original model and quantized model.

**Lemma D.1.** *Let*

$$error(\hat{w}) = \langle g_w, \hat{w} - w \rangle + (\hat{w} - w)^T H_w (\hat{w} - w) + \cdots$$

*be the Taylor series expansion of the KL divergence where $g_w$ is the gradient and $H_w$ is the Hessian. Then*

*1. $g_w = 0$,*

*2. $H_w = \mathbb{E}_{z \sim \mathcal{D}_{data}} \mathbb{E}_i \mathbb{E}_{t \sim p_w(\cdot|z_{<i})} [(\nabla_w \log p_w(t|z_{<i}))(\nabla_w \log p_w(t|z_{<i}))^T]$*

*Therefore $error(\hat{w}) \approx \mathbb{E}_{z \sim \mathcal{D}_{data}} \mathbb{E}_i \mathbb{E}_{t \sim p_w(\cdot|z_{<i})} [\langle \nabla_w p_w(t|z_{<i}), \hat{w} - w \rangle^2]$.*

*Proof.* To simplify notation, we will ignore the $z, i$ variables coming from $\mathbb{E}_{z \sim \mathcal{D}_{\mathrm{data}}}$ and $\mathbb{E}_i$ and also drop them from $p_w(\cdot|z_{<i})$ and just write $p_w(\cdot)$. Adding these back and taking expectations over these variables, we get the desired result. We can expand the KL divergence using Taylor series and evaluate the first and second order terms.

$$\begin{aligned}
\mathrm{error}(\hat{w}) &= D_{KL} \left( p_w(\cdot) \,\|\, p_{\hat{w}}(\cdot) \right) \\
&= -E_{t \sim} \left[ \log p_{\hat{w}}(t) - \log p_w(t) \right] \\
&= -E_{t \sim p_w} \left[ \langle \nabla_w \log p_w(t), \hat{w} - w \rangle + (\hat{w} - w)^T \nabla_w^2 \log p_w(t)(\hat{w} - w) + \cdots \right] \\
&= \langle g_w, \hat{w} - w \rangle + (\hat{w} - w)^T H_w (\hat{w} - w) + \cdots
\end{aligned}$$

where $g_w = -E_{t \sim p_w}[\nabla_w \log p_w(t)]$ and $H_w = -E_{t \sim p_w}[\nabla_w^2 \log p_w(t)]$.
(1) We first evaluate $g_w$.

$$\begin{aligned}
g_w = -E_{t \sim p_w}[\nabla_w \log p_w(t)] = \mathbb{E}_{t \sim p_w} \left[ \frac{\nabla_w p_w(t)}{p_w(t)} \right] \\
= \sum_t \nabla_w p_w(t) \\
= \nabla_w (\sum_t p_w(t)) \\
= \nabla_w(1) = 0.
\end{aligned}$$

(2) We now evaluate $H_w$.

$$
\begin{aligned}
H_w &= -E_{t \sim p_w}[\nabla_w^2 \log p_w(t)] \\
&= -\mathbb{E}_{t \sim p_w}\left[\nabla_w\left(\frac{\nabla_w p_w(t)}{p_w(t)}\right)\right] \\
&= -\mathbb{E}_{t \sim p_w}\left[\frac{\nabla_w^2 p_w(t)}{p_w(t)} - \frac{(\nabla_w p_w(t))(\nabla_w p_w(t))^T}{p_w(t)^2}\right] \\
&= -\mathbb{E}_{t \sim p_w}\left[\frac{\nabla_w^2 p_w(t)}{p_w(t)} - (\nabla_w \log p_w(t))(\nabla_w \log p_w(t))^T\right] \\
&= -\sum_t \nabla_w^2 p_w(t) + \mathbb{E}_{t \sim p_w}\left[(\nabla_w \log p_w(t))(\nabla_w \log p_w(t))^T\right] \\
&= -\nabla_w^2\left(\sum_t p_w(t)\right) + \mathbb{E}_{t \sim p_w}\left[(\nabla_w \log p_w(t))(\nabla_w \log p_w(t))^T\right] \\
&= -\nabla_w^2(1) + \mathbb{E}_{t \sim p_w}\left[(\nabla_w \log p_w(t))(\nabla_w \log p_w(t))^T\right] \\
&= \mathbb{E}_{t \sim p_w}\left[(\nabla_w \log p_w(t))(\nabla_w \log p_w(t))^T\right]
\end{aligned}
$$

$\square$

