# OpenReview forum: "DiscQuant: A Quantization Method for Neural Networks Inspired by Discrepancy Theory"
_ICLR.cc/2025/Conference — Submitted to ICLR 2025_

### Official Review · Reviewer_ney3 · 2024-10-27

**Soundness:** 3
**Presentation:** 3
**Contribution:** 3
**Rating:** 6
**Confidence:** 2

**Summary:**

This paper proposes DiscQuant, a data-driven rounding algorithm for post-training quantization (PTQ) of large neural networks. DiscQuant assumes that the gradients of the original model are low-rank. Under this assumption, the authors prove that their algorithm can arbitrarily minimize the upper bound on the error of the quantized model, given an accordingly large number of samples from the target data distribution.

The algorithm is primarily concerned with the second (rounding) step in quantization. The rounding process aims to minimize the KL divergence between the distributions of the next token predictions of the original and quantized models. The proposed algorithm significantly improves against the existing state-of-the-art when quantizing LLMs like Phi-3-mini-4k-instruct and Meta-Llama-3.1-8B-Instruct.

**Strengths:**

1. The paper is well-written and provides sufficient theoretical results using discrepancy theory (§3).
2. The proposed algorithm is agnostic to the quantization grid, which makes it quite generally applicable.

**Weaknesses:**

While I acknowledge the contributions made in this work, I hesitate to place a higher score here because of the following reasons.

1. The scope of the currently presented applications is limited to LLMs. I feel that if VLMs (e.g., Llava-v1.6-34b) were to be tested, consistent results there would greatly increase the paper's relevance.
2. Even within LLMs, the tested models seem quite small. Quantization is concerned with memory efficiency, so it makes sense to perform experiments with large models (e.g., Llama-3.1-70B-Instruct) that would pose greater storage challenges than the ones tested in the current work. Understandably, as the authors note (lines 405-406), DiscQuant requires two copies of the model to be stored in memory during the quantization process. This limits the scope for large-scale experiments in academic settings, but it also highlights a major shortcoming of the approach.
3. The method needs access to data, which might be the bottleneck for many practitioners who, for instance, quickly need to prototype a handful of models but don't have the resources to run the full models, neither the data to quantize them using a DiscQuant-like algorithm. Besides, as the authors note in the last paragraph, the choice of data is in itself non-trivial.

**Minor issues**

1. The authors may want to confine the abstract to one paragraph in the interest of adherence to the ICLR 2025 guidelines.
2. Line 133: let's --> lets?
3. Line 379: close *to* the polytype?

**Questions:**

It would be great to hear the authors' comments on the points raised above (see **Weaknesses**). In addition,

1. Lines 249-251: "We make this assumption because we don’t want to change any parameter of the original model too much during quantization, consider it an important property of algorithms we design" -- could the authors further explain this design choice, and the drawbacks of potentially relaxing this restriction?
2. do the authors plan to release their code?

---

> ### Author Response · Authors · 2024-11-22
> **Author response**
>
> **Q. Why not VLMs**
>
> **A.** We believe that experiments on vision models are beyond the reasonable scope of this paper. To the best of our knowledge, other major works on PTQ (GPTQ, QuIP, OmniQuant, etc) focus on LLMs only. However, our method is general and can be used on CNNs or vision transformers.
>
> **Q. Experiments on Larger LLMs**
>
> **A.** We have initiated experiments on Llama-3-70B, but given our limited resources we cannot finish the experiments (especially hyperparameter tuning) in time for the rebuttal. We will update the paper with results as soon as we can get them.
>
> **Q. On Data and resources**
>
> **A.** In our experiments we only use several thousand samples from an open source dataset called RedPajama which should make our method accessible to any practitioner. Our experiments in Section 5.3 were to show the improvement you could get from tailoring your quantization data to your task and even in those experiments we will only use a few thousand samples from the end task. So we believe that not having access to data is not a valid criticism of our method.
>
> On the matter of compute resources, our method requires storing two copies of the model and doing backprop on one of them similar to distillation. This will require large memory and especially for extremely large models like Llama 70b or 405b this will require several GPUs. But we believe that PTQ methods should be evaluated foremost on the quality of the quantization rather than the compute resources needed. A company/lab which pretrains a model and wants to create a quantized version of that model for cheaper inference cares more about the quality of quantization rather than resources needed for quantization. In any case, the amount of resources needed for quantization using our method or most PTQ methods is negligible compared to the pretraining costs. Therefore we appeal to the reviewer to not see this as a serious shortcoming.
>
> **Q. Why only search up/down in the quantization grid.**
>
> **A.** Great question! So many resources are spent on pretraining and aligning modern LLMs. We do not want our quantization method to change the original model too much, especially because it is common to use only a couple thousand samples for calibration during PTQ. By only searching up/down in the quantization grid, we are ensuring that trillions of tokens worth of training is not overwritten by several thousand tokens worth of post training quantization. Relaxing this design choice may degrade the LLM in ways not captured by the relatively simple benchmarks that are commonly used to evaluate quantized models.
>
> **Q. Will you release code?**
>
> **A.** Yes!

---

> ### Comment · Reviewer_ney3 · 2024-11-28
> **Response to authors**
>
> Thanks to the authors for their efforts in responding! Thanks for conducting Llama-70B experiments as well. My apologies for the delayed response.
>
> Regarding data and resources: my original comment was largely concerned with data, but I believe that Reviewer 7uqc expresses a similar concern more effectively, saying, "it is not that fair to claim the superior performance of DiscQ from comparisons with PTQ methods, provided that the costs are much higher than those PTQ methods". I am curious to see the authors' clarification on this.
>
> Resource efficiency is critical, and the argument that this is insignificant compared to what's spent in pre-training, seems, in my opinion, to unfairly downplay the issue. Regardless of availability, one would generally seek to minimize resource expenditure, if doing so does not degrade the outcome significantly. Beyond that, there also lies the matter of complexity: the more complex a method is (simply in terms of running it, for example), the less significant its performance gains become and the harder it is to debug. Consider the very general case (that still applies here) where one needs to set up a third-party codebase and adapt it to their application. Given two third-party implementations with similar resource expenditure, one would generally use another tie-breaking criterion such as simplicity or ultimate performance gain: this would be done to justify the expenditure of an engineer's time, which is arguably a more expensive resource than compute. This is why it becomes so important to compare with methods that use similar resources.
>
> Given the currently available information, I will maintain my score of acceptance but with reservations.

---

### Official Review · Reviewer_7uqc · 2024-10-31

**Soundness:** 3
**Presentation:** 3
**Contribution:** 2
**Rating:** 3
**Confidence:** 5

**Summary:**

The paper studied the generalization gap of quantization using techniques from discrepancy theory under the assumption that the gradient is approximately low-rank. Based on theoretical analysis, the authors proposed a new quantization algorithm, named DiscQuant. Experiments are conducted to compare the proposed algorithm with existing quantization algorithm, like RTN and GPTQ.

**Strengths:**

1. Theoretical analysis is solid. The paper provides a solid theoretical analysis to study the generalization error of quantization.
2. Connection with discrepancy theory. It is novel to apply techniques of discrepancy theory to neural network quantization.

**Weaknesses:**

The weaknesses of the paper mainly come from the numerical algorithm and experiments.

1. The proposed algorithm solves the optimization problem (3). It seems like full-model training is required to solve the problem (3), which can be too expensive for state-of-the-art large language models.
2. The authors only compare the proposed approach with RTN and GPTQ, which are relatively old PTQ algorithms in the area. The results seem to show a big gap between the SOTA PTQ methods.

**Questions:**

1. The result in the main Theorem 3.3 only depends on the data distribution. In practice, the PTQ performance also depends on the value distribution of weight matrices. If values in weights are evenly distributed, the performance after PTQ is usually better. The authors also mentioned incoherence processing, a popular method to reduce the ranges of weights. I am very curious why the distribution of weights is not reflected in the main theorem 3.3 for generalization error.
2. Could authors explain more about how to solve the problem (3) in their algorithm? In my understanding, we may need to run backpropagation and optimize the problem (3). If that is the case, the cost is almost the same as training the full model which is too much for PTQ compared to other existing algorithms. By using the resources, people can directly run distillation or quantization-aware training for better compression and better performance.
3. A follow-up question, can authors provide time and memory costs for the proposed algorithm?
4. Nowadays, there are lots of new algorithms for quantizing neural networks. It is better if authors can compare their algorithm with those works. Some new algorithms are listed in the following:
      * Zhang, Aozhong, et al. "MagR: Weight Magnitude Reduction for Enhancing Post-Training Quantization." arXiv preprint arXiv:2406.00800 (2024).
      * Shao, Wenqi, et al. "Omniquant: Omnidirectionally calibrated quantization for large language models." arXiv preprint arXiv:2308.13137 (2023).
      * Chee, Jerry, et al. "Quip: 2-bit quantization of large language models with guarantees." Advances in Neural Information Processing Systems 36 (2024).
      * Liu, Zechun, et al. "SpinQuant--LLM quantization with learned rotations." arXiv preprint arXiv:2405.16406 (2024).
The results shown in the paper seem to be worse than the reported results in these most recent algorithms. It will be great the authors can choose some of them to make comparisons and explain the performance gap.

---

> ### Author Response · Authors · 2024-11-22
> **Author response**
>
> **Q.  “Why can’t you directly run distillation or quantization-aware training for better compression and better performance”**
>
> **A.** Thanks for asking us to clarify. Our method has similar memory requirements as distillation, where we hold two copies of the model in memory. And like distillation, we optimize one copy of the weights. However, our method uses only a couple thousand samples, which is negligible compared to what is required for distillation or quantization aware training or full model training which require training again on a significant portion of the pretraining data. We believe that the statement “the cost is almost the same as training the full model” needs this additional context. Though per iteration costs are similar, our method requires way less samples and iterations and therefore not comparable to these methods. And while our method requires greater computational resources than GPTQ, we have shown that it achieves superior compression. So we do not believe that our method should be summarily rejected as a PTQ method because of its computational requirements. Ultimately, when a company which trained a model wants to create a quantized version of their model, they do have the resources to apply any method to quantize it. What matters more in this case is the quality of the quantized model rather than the amount of resources needed to quantize it. In any case, the resources needed for our method and most PTQ methods is negligible compared to pretraining.
>
> **Q. Comparison with SOTA PTQ methods**
>
> **A.** Thanks for asking. We ran the recent SOTA OmniQuant [1] method in our experimental setting: Llama-3-8B-Instruct on the Wikitext, PIQA, and HellaSwag tasks. We run the OmniQuant method from their official github, using their default parameters for weight-only quantization. We add the `--symmetric` argument to match the symmetric quantization grid used in our experiments. We see that our method DiscQuant beats OmniQuant on all tasks and at all bit settings.
>
> Also, we do compare against QuIP, see Section 5.2.
>
> | Method | Wbits | Wiki$\downarrow$      | PIQA$\uparrow$ | Hella$\uparrow$ |
> |------------|---------|-----------|---------|---------|
> | -----------| 16.0   | 8.7       | 81.3   | 79.3  |
> | RTN      | 3.0     | $4.4$E$3$ | 52.4 | 29.1 |
> | GPTQ   | 3.0     | 23.2     | 66.6   | 45.8  |
> | OmniQ  | 3.0     | 315      | 55.7   | 31.9  |
> | DiscQ   | 3.0     | 15.       | 73.2   | 64.4  |
> | RTN     | 3.25    | 15.2     | 75.2  | 71.4  |
> | GPTQ   | 3.25   | 10.7     | 76.7  | 74.4  |
> | OmniQ | 3.25    | 14.6     | 74.2  | 69.1  |
> | DiscQ   | 3.25   | 10.5     | 79.1   | 75.1  |
> | RTN      | 3.5     | 12.7      | 76.7 | 73.0 |
> | GPTQ   | 3.5    | 10.4      | 77.3  | 75.1 |
> | OmniQ | 3.5     | 12.7      | 76.4  | 72.8 |
> | DiscQ   | 3.5     | 10.3      | 79.2  | 76.3 |
> | RTN      | 4.0    | 12.5      | 77.6  | 74.7 |
> | GPTQ   | 4.0    | 9.9        | 78.4  | 75.9 |
> | OmniQ  | 4.0    | 10.9      | 78.1  | 76.3 |
> | DiscQ    | 4.0   | 9.8        | 79.2  | 76.9 |
> | RTN      | 4.25  | 9.4       | 80.1   | 78.0 |
> | GPTQ   | 4.25   | 9.1       | 79.6  | 77.9 |
> | OmniQ  | 4.25   | 9.4      | 79.9   | 78.1 |
> | DiscQ   | 4.25   | 9.1       | 79.9  | 78.4  |
> | RTN      | 4.5    | 9.3       | 80.3  | 78.4  |
> | GPTQ   | 4.5    | 9.0       | 79.6   | 78.1 |
> | OmniQ | 4.5    | 9.3        | 80.1   | 78.3 |
> | DiscQ   | 4.5     | 9.1       | 80.6   | 78.7 |
>
>
> [1] Shao, Wenqi, et al. "Omniquant: Omnidirectionally calibrated quantization for large language models." arXiv preprint arXiv:2308.13137 (2023).
>
>
> **Q.Thm 3.3 weight distribution**
>
> **A.** Great question! As you rightly pointed out, the quantization error will depend on the distribution of weights as well. In our modeling, we made the simplifying assumption that the quantization grid is uniform and fine enough to always give a plus-minus delta approximation to each true weight. Therefore the quantization error depends on delta as well. See the equation in Section 3.1 for this which states that quantization error $\approx \delta \langle \nabla_w f(w;s),x-y \rangle$. In Theorem 3.3, we ignored this delta factor since it is a constant and only focused on the $\langle\nabla_w f(w;s),x-y\rangle$ part. In Appendix C, we have clarified what happens when the quantization grid is not uniform, in this case, as you pointed out, the quantization error will also depend on the distribution of weights since the matrix $M$ will have gradients scaled by $(w^{up}-w^{down})$. In this case, the generalization bound in Theorem 3.3 will need to assume that the covariance of these scaled gradients need to have decaying eigenvalues and the exact theorem continues to hold. If you think this is worth explaining in detail, we are happy to add this theorem in the appendix and write a footnote in the main body.

---

> > ### Comment · Reviewer_7uqc · 2024-11-26
> >
> > I would like to thank the authors for taking the time to respond to my concerns. Here are some comments regarding the responses.
> >
> > 1. Thanks for the explanation! I understand that the computational costs of both PTQ and the proposed algorithm are negligible compared to the full pre-training of neural networks. But, the proposed DiscQ still uses much more resources than PTQ, where the peak cost of DiscQ should be the same as that of QAT and distillation. It is considerably expensive especially when scaling up to large models.
> >
> > It is okay for me that a quantization algorithm uses lots of resources. But, it is not very clear to me whether the performance improvement comes from the proposed algorithm or from the increased computation. I think that if we simply run QAT and distillation on some small calibration data either after PTQ or without PTQ, the quantization performance may also improve. Hence, it is not that fair to claim the superior performance of DiscQ from comparisons with PTQ methods, provided that the costs are much higher than those PTQ methods. It might be better to also compare DiscQ with methods like QAT with distillation, that use similar resources to further demonstrate the performance of proposed algorithm.
> >
> > 2. I just noticed that the experiments are done on Llama-3-8B-Instruct model. Just wondering why do you choose this model? It seems most papers on quantization use base models for experiments. If the paper uses the same model, there can be many results in existing papers for comparisons.
> >
> > 3. Thanks a lot for the explanations on weight distribution. It will be good to add that theorem to the paper.

---

> > > ### Author Response · Authors · 2024-12-03
> > > **Author response (2)**
> > >
> > > **Q. On computational resources and a fair comparison**
> > >
> > > **A.** We hypothesize that our method is superior over methods like GPTQ because it optimizes the entire model simultaneously; in contrast GPTQ only optimizes layer-wise. Our model is computationally more expensive than GPTQ, but this greater expense is due to the structure of our algorithm. So we believe the question to “[does the] performance improvement come from the proposed algorithm or from the increased computation”, is both. As a counterpoint, it is not obvious how to naturally extend GPTQ with greater computation in a way that would make it superior to our method. We tuned GPTQ with 1k, 4k, and 8k samples; 8k samples were not consistently better.
> > >
> > > Can the reviewer provide references to QAT or distillation papers which use several thousand samples and are effective at quantization? We understand the point you are making, but do not know of any papers that claim to do this.
> > >
> > > **Q. Why Instruct model**
> > >
> > > **A.**
> > > We did not have a strong reason for using the instruct models. However we caution against comparing results between papers without at least verifying the full precision baselines or better, evaluating code details such as package versions or prompting details. These comparison issues come up frequently in our experience.

---

### Official Review · Reviewer_6GRV · 2024-11-01

**Soundness:** 3
**Presentation:** 3
**Contribution:** 4
**Rating:** 6
**Confidence:** 3

**Summary:**

This paper proposes "DiscQuant", a method to quantize weights of a large language model, which aims to improve the quantization-accuracy trade-off. The authors argue that quantization can be roughly thought of as two steps: 1) coming up with a good quantization grid,  2) a good rounding scheme that maps the exact parameters to a point in the discrete grid. Authors argue that while there ha been much effort on step 1, there is less focus on step 2, which is the rounding procedure. The key theoretical ingredient of the paper is to formulate the problem of rounding in the framework of discrepancy theory. In particular, they aim to bound the errors introduced due to rounding for all seen and unseen samples

After formulating the problem in an exact manner, which is roughly the KL divergence between unrounded and rounded model predictions, they go on to make several assumptions: 1) they assume that a simple rounding up/down suffices, and there is no needs for "jumps" in the rounding, which becomes more reasonable if the grid is fine enough. 2) they assuming a particular low-rank structure about the weight gradients, and that gradients are well behaved (defined as $\beta$-reasonable). This is key to achieve an unseen or generalization error bound. 3) They assume that  the first order approximation to the error is sufficient for calculating the errors. Here, they stress the fact that while loss gradients averaged over samples may be small, the per sample gradients are not small and in fact dominate the errors due to rounding.

After making thee assumptions, the paper goes on to state the main theorem of the paper, Theorem 3.3, which gives the guarantee for generalization error, and subsequently introduce the algorithm that finds such a rounding efficiently in section 4. Finally, the paper presents, what seems like very compelling evidence that their proposed quantization scheme outperforms two baselines (RTN and GPTQ, see Figure 2). They also empirically test two key assumptions, the first order error approximation in Figure 3, and the low-rank structure of gradients in Figure 4,  which substantiates why their theoretical results are applicable.

**Strengths:**

In general, I find this work to be very strong, as it works on an important practical problem, and presents an innovative appraoch that is both theoretically grounded and is practically impactful. Let me enumerate these strengths one by one:
- The recognition of the problem with exiting quantization methods, in that they ignore the importance of the rounding step and only focus on the quantization grid, seems to be a highly relevant and important message of this paper
- After recognizing this problem, authors propose a very nice and innovative approach by formulating the problem in terms of discrepancy theory, which in hindsight, seems like an excellent choice for this problem
- The assumptions necessary for the theory, seems to be well substantiated and reasonable, and authors go to reasonable length to explain and justify them, rather than hiding them.
- The empirical results of the paper are equally strong as the theoretical results.

**Weaknesses:**

My main criticism of the current draft is its lack of clarity on some of the technical/theoretical parts of the paper.
For example, the paper would benefit a lot from an expanded explanation of the basics, ie, the basic discrepancy theory setup that they are casting their problem to, explaining the e Lovett-Meka algorithm that they are invoking so many times in detail, and then explaining what problems (complexity perhaps) it has, and what they do to fix it. Currently, it seems like the paper assumes the reader is already familiar with all thee topics and they only present bits that are novel. For reference, I spent nearly 1 hour trying to catch up with thee basics, namely the  Lovett-Meka algorithm,  but still only partially understood the technical details of the paper.

Even after an expanded explanation on the theory basics, I think the paper needs to give more intuitive high level view of the algorithm. In particular, in section 4, there could be more explanations on what is the idea behind this formulation/heuristic of minimizing along arbitrary direction $c$? Perhaps a geometric intuition, similar to the one given in Fig 1, could be given here?

Another point is the lack of clarity on the complexity of the proposed approach. From my limited understanding, the Lovett-Meka algorithm involves iteratively solving a SDP, which could be highly expensive in some cases. It sounds like this paper addresses some of these complexity issues via the heuristics (eg, lines 371-272). But it's hard to fully understand the solution, if the reader hasn't fully understood the problem. So an expanded section on background methods and their complexity, and then the complexity of the heuristics-based approach, would help the reader quite a bit.

**Questions:**

One of my biggest questions that remained unresolved while reading the paper was, when discussing gradient covariance matrix, are we talking about the covariance within a fully connected layer, or could it be between different layers? In general,  there is some ambiguity as to what “n” in the parameter space really entails here. Is it the full parameter space (all parameters together), or is this procedure applied per each fully connected layer? If it is the latter, the next question is, in which order are these soundings applied? And would the authors re-calculated the gradients after each step?

On a related note, if this procedure is done iteratively for different fully connected layers, can authors expand on this? For example, do they quantize every layer as if other layers are in the original form, or do they have an iterative approach where the the quantization of first layer impacts the quantization of the subsequent layers.

If this is done in one shot, is that a complex operation? In general, I would also highly welcome some notes on complexity of running this algorithm.

---

> ### Author Response · Authors · 2024-11-22
> **Author response**
>
> **Q. Expanding on  discrepancy theory; more intuition on the algorithm, complexity of Lovett Meka.**
>
> **A.**.We regret that we couldn’t provide a more detailed introduction and connections to discrepancy theory in the main body of the paper due to lack of space. We explained the connection briefly in Section 3.1. To recall, $M$ is the matrix whose rows are per-sample gradients and we are given some fractional $y\in [0,1]^n$ which corresponds to the true weights. Our goal is to find an $x\in \\{0,1\\}^n$ such that $M(x-y)\approx 0$ (which corresponds to making the first order approximation of quantization error zero). A famous result of Lovasz et al. 1986 (cited in Section 3.1) shows that
>
> $$\min_{x \in \\{0,1\\}^n} ||M(x-y)||_{\infty} \le herdisc(M)$$
>
> where $herdisc(M)$ is the hereditary discrepancy of M defined as
>
> $herdisc(M)=\max_{S \subset [n]} \min_{z\in \\{-1,1\\}^S} ||M_S z||_{\infty}$
>
> where $M_S$ is the submatrix of $M$ formed by columns $S\subset [n]$. And there are many well-known techniques to bound herdisc(M). We didn’t go into details because of lack of space. But if the reviewers think it is useful, we can add an appendix elaborating on these connections  and also give a gentle introduction to discrepancy theory.
>
>
> One important distinction concerns separating our analysis from our method. Discrepancy theory, including the Lovett-Meka algorithm, was used to obtain a generalization bound and obtain novel insights on the quantization problem.  It just proves mathematically that low-rank gradient structure can imply existence of good quantization. However, we do not use Lovett-Meka in our actual method as it is very expensive to run as the reviewer correctly pointed out. It involves solving a Gram-Schmidt orthonormalization process at each step of the algorithm which is infeasible when the dimension n is in the order of billions such as in LLMs. Therefore our actual method DiscQuant minimizes KL divergence and a linear regularization term using a standard stochastic gradient descent approach. We explained why this is morally similar to finding a vertex of the polytope in Section 4. Recall from Fig 1 that our rounding solution lies on a vertex of polytope “K”. In order to find any vertex, we can maximize $\langle c,x \rangle$ such that $x \in K$, for any direction $c$. It is a well-known fact in linear programming that a linear function is always maximized or minimized at a vertex of a polytope. Intuitively, we can see this by imagining placing a ball inside a convex polytope and let it fall under the action of gravity, it will eventually stop at a vertex of the polytope (if the orientation of the polytope is randomly chosen which is equivalent to choosing a random direction $c$, then almost surely it will not have any horizontal surface where the ball can rest).
>
> Our final algorithm DiscQuant just uses standard (projected) stochastic gradient descent to minimize the objective (3) in Section 4, so it only requires computation of average gradient on a batch of samples for each step using standard backprop. It does a descent step followed by a projection step onto the hypercube which can be achieved by clamping the x parameters to [0,1].
>
> **Q. Is gradient covariance within a layer, or for the entire model?**
>
> **A.** We are discussing the gradient covariance with respect to the entire model and we use n to denote all the parameters of the model. Note that our method does not compute this covariance matrix explicitly; we reason about this matrix in our analysis to prove the generalization bound. As explained in the top of Section 5, our method has similar memory requirements as knowledge distillation, which also requires holding two copies of the model in memory. We perform backpropagation and optimize one copy of the model weights. However our method only uses a few thousand samples, far fewer than pretraining, distillation, or quantization aware training.

---

> ### Comment · Reviewer_6GRV · 2024-11-22
>
> I thank the authors for the clarifications and responses. Some further thoughts:
>
> While I appreciate the explanations here, what I was hoping for was a more intuitive understanding of the method. Typically, readers who aren't familiar with the literature of a paper, can only "compile" the theoretical statements, but cannot gain insights into the theories. These insights will be crucial in how they will apply the method, and diagnose it if it doesn't work quite well. In that sense, the explanations provided here don't really address my concern on clarity of the presentation.
>
> Some suggestions:
> - Can authors provide a toy example (either here or in the paper) of how DiscQuant works? If possible, that would go a long way in clearing up some details?
> - Can this toy example, or some other example, explain intuitively why the low-rank assumption helps better quantization?
> - Can authors write a pseudo-code for DiscQuant, with clear explanations for all the terms there (rather than a vague "n" term, mention it represents all parameters, or what $c$ means, etc). Also can they provide a complexity estimate?
>
> follow-ups on low-rank:
> - Since authors clarify $n$ denotes all parameters of the model, does the low-rank structure somewhat arise from the fact that interactions between different components  is very small compared to interactions within a module? For example, parameters within a fully connected layer have a lot of interaction, as opposed to two different layers. Is this the type of low-rank structure that is needed here? Or should the low rank go beyond this, and even the gradients within a fully connected layer must be low-rank?
> -  I read the discussion in response to reviewer `TWTt`, but I am still unsure what authors are implying here. Are authors suggesting low-rank assumption is necessary for theoretical analysis, or is it actually needed in practice too? In other words, if it's violated, is it likely that DiscQuant (or other methods) fail in practice too?
>
> Second follow-up: Since the "method" is not really LM algorithm, can they clarify if the main Theorems presented are for LM algorithm, or for the approximate rounding technique used in DiscQuant? The language of the theorems " ... there is a polynomial time algorithm ... " is very vague and leaves a lot of room for interpretation here. The subsequent section 4 does not clarify this either.

---

> ### Author Response · Authors · 2024-11-25
> **Author Response (2)**
>
> **Q. A more intuitive understanding of the method**
>
> **A.**. We will include a more approachable explanation for our method in the paper. We believe that the geometric interpretation in Figure 1 provides the best intuition. Our introduction provides a thorough but perhaps too technical explanation of this intuition, but we will try and restate it here.
>
> We reparametrize the model weights as $w^x =(1-x)\cdot w^{down}+x\cdot w^{up}$ by interpolating between the rounded down weights $w^{down}$ and rounded up weights $w^{up}$. We now consider $x$ as the trainable parameters which are constrained to the hypercube $H=[0,1]^n$ as in Figure 1. A full rounding corresponds to a vertex of the hypercube $x\in \\{0,1\\}^n$. If $x_i=0$, then the $i^{th}$ parameter $w_i$ is rounded down and if $x_i=1$, then the $i^{th}$ parameter is rounded up. Our problem therefore is to push all optimization parameters to either $0$ or $1$, corresponding to either rounding down or up to the nearest points on the quantization grid.
>
> But any rounding will not do; we want to ensure that our rounding does not degrade the model. By looking at the first order taylor approximation of the rounding error, this can be written as a set of linear constraints on $x$. This constraint corresponds to the hyperplane $V$ in Figure 1. So in order to round the parameters and have low model degradation, we need to find points of $V$ intersected with the hypercube $H$ with many coordinates set to 0 or 1. This corresponds to finding vertices of the polytope $K= V \cap H$, shown as red points in Figure 1. If the dimension of $V$ is $n-m$, then these vertices have $n-m$ values set to 0 or 1 and at most $m$ values which are still fractional. So we want $m$ to be small to find a nearly full rounding. Now this is where the low-rank assumption comes in. The $m$ here is approximately the rank of the gradient space, so low rank implies an almost fully rounded solution with low rounding error.
>
> Now how does DiscQuant solve this problem. Firstly we can’t even store the subspace $V$ explicitly since storing it will require storing an $m \times n$ matrix where $n$ is the number of parameters and $m$ is a few thousand. Therefore we instead solve:
> $$\min_x D_{KL}(f(w; s) || f(w^x;s)).$$
> The set of $x$ which minimize this objective roughly corresponds to the subspace $V$ as shown in the equations in Section 4. Now our goal is to find
> $$\min_{x\in \\{0,1\\}^n} D_{KL}(f(w; s) || f(w^x;s)).$$
> This is a discrete optimization problem, so we need to relax it to a continuous optimization problem.
> $$\min_{x\in [0,1]^n} D_{KL}(f(w; s) || f(w^x;s)).$$
> But the solution to this is trivially zero which happens when $w^x=w$, i.e., no rounding has actually taken place. So add a linear penalty term which forces each $x_i$ to be either $0$ or $1$. We have already given intuition in our previous response why optimizing a linear penalty over a convex polytope leads us to a vertex. Therefore the final objective is:
>
> $$\min_{x\in [0,1]^n} L(x;s)=D_{KL}(f(w; s) || f(w^x;s)) +\lambda \langle c , x \rangle.$$
>
> Finally, we choose $c$ in a special way such that vertices of the hypercube which are closest to the original weights are preferred (last equation in Section 4). So we are not repeating this simple calculation here.
>
> We solve this objective using projected stochastic gradient descent as follows:
> DiscQuant Psuedocode:
>
> ---
> For $t=1$ to $T$:
> - Sample $s_1,s_2,\dots,s_B\sim D_{data}$ from the calibration data
> - $g=\frac{1}{B} \sum_{i=1}^B \nabla_x L(x;s_i)$
> - $x \leftarrow x- \eta g$ #descent step
> - $x \leftarrow clamp(x,0,1)$ #project to hypercube
>
> $\hat{x}\leftarrow round(x)$ #At this point, most $x_i$ should be close to $0$ or $1$. Round whatever is left to $0$ or $1$ greedily.
>
> Output $\hat{w}=w^{\hat{x}}$ as the quantized weights
>
> ---
>
> Complexity: Each iteration takes linear time in the number of parameters $n$ and batch size $B$ and quadratic in the context length $L$ (for transformer architecture). The projection step is just clamping, so linear in $n$. Therefore total running time is $O(nBL^2T)$. This is exactly the same as usual stochastic gradient descent for standard training of transformer networks. The only extra work is to also evaluate the logits of the original model (using only the forward pass) which is a small fraction of the total cost, so will not change the asymptotics.

---

> > ### Author Response · Authors · 2024-11-25
> > **Author Reponse (3)**
> >
> > **Q. Low-rank followup**
> >
> > **A.** As pointed out in the reply to reviewer TWTt, low-rank of gradient space is necessary for any PTQ method to work (not just ours). If the gradient space is isotropic (i.e., the covariance matrix of gradients is approximately the identity matrix), then we can prove that any PTQ method will have to incur a large rounding error (both in theory and in practice). The only way to avoid this would be to do an extensive Quantization Aware Training on a large portion of the pretraining dataset which requires that the model weights will change significantly compared to the pretrained weights. So the low-rank assumption is not just a convenient assumption for our theoretical analysis, it is a necessary condition for PTQ methods to work in practice as well. And we empirically check that this condition is met by LLMs in our experiments.
> >
> > Regarding the structure of low-rankness. Low-rankness of entire model gradients in particular implies even if you look at the gradients w.r.t parameters inside a single linear layer, these should also be low-rank. So it is not just about interactions across different layers. Low-rankness of model gradients means that there are a few important orthogonal directions along which model loss function decreases or increases singficantly and most other orthogonal directions the loss function is flat.
> >
> > **Q. Is the generalization theorem for Lovett-Meka or DiscQuant**
> >
> > **A.** The generalization theorem shows that there is a polynomial time algorithm which can find a good rounding (the polynomial time algorithm is based on Lovett-Meka). But the run-time complexity of this algorithm is at least cubic in the number of parameters and therefore it cannot be implemented in practice.
> >
> > We therefore give a heuristic algorithm called DiscQuant inspired by our theoretical insights. The generalization bounds we proved do not apply to this heuristic algorithm. We only give experimental evidence to show that it actually works in practice and beats GPTQ which is the prior SOTA method for rounding. We will make this point clearer by updating our paper and state the generalization bounds do not apply to DiscQuant.
> >
> > The purpose of the generalization bound is to state that there exists a good rounding under the low-rank assumption and it can be found in polynomial time. It is in similar spirit to Khachiyan’s famous ellipsoid algorithm for solving linear programming which is a polynomial time algorithm. It shows that it is possible to solve it in polynomial time, but people don’t use it in practice. Instead they use heuristic algorithms like the Simplex algorithm (which cannot be proven to converge in polynomial time).

---

### Official Review · Reviewer_TWTt · 2024-11-07

**Soundness:** 3
**Presentation:** 3
**Contribution:** 2
**Rating:** 3
**Confidence:** 4

**Summary:**

This paper introduces a new method of quantizing neural networks with inspiration from theory of discrepancies. Traditionally, quantization in neural networks is composed of two processes, namely defining a grid of quantization and rounding model weights to match that particular grid. Standard rounding procedures are typically RTN, alongside data-dependent methods, some of which include GPTQ; however, in this paper, the round the weights using the theory of discrepancies approach without resulting in an increase in the loss on unseen data.

DiscQuant relies on a mathematical framework to guarantee low error through rounding of nearly all model weights based on a low-rank assumption of gradient covariance. The authors establish theoretical bounds that their method can be guaranteed to achieve the expected generalization error to be at most epsilon on the data distribution, conditioned on certain low-rank conditions being met in the gradient space. They use such theoretical results to design a practical rounding algorithm that rounds the model weights from such an optimizer in a way that minimizes a regularized objective function combining KL divergence and linear constraints; it thus preserves overall model performance.

Extensive experiments on Phi-3-mini-3.8B and Meta-Llama-3.1-8B models across tasks and quantization levels indicate the superior performance of DiscQuant against RTN and GPTQ specifically at low bits. The authors are able to show that DiscQuant has gained in performance over benchmarks GSM8k, ARC Challenge, and PIQA, which then comes to expose its generalizability and robustness over any possible format of quantizations such as block scaling and incoherence processing.

**Strengths:**

This paper contributes a new input to the field of neural network quantization by pursuing a novel angle inspired by discrepancy theory in tackling the rounding problem. Traditional methods of quantization lie on two strands: Either the quantization grid must be designed efficiently, or the standard rounding technique, Round-to-Nearest, must be used. In its work, this paper introduces discrepancy theory for optimizing the rounding step and shows it can get high performance with all weights rounded to almost all possible bits but with a rather small approximation error. This result is established in a theoretical framework that automatically ensures low error, such as in the low-rank covariance matrices of the gradients, and so on. Therefore, this problem formulates a relatively little-investigated area of quantization in a new way.

The quality of this paper is very good in terms of methodology, with sound theoretical underpinnings for the proposed DiscQuant method to work. The authors systematically derive bounds on generalization error, effectively coupling their method with gradient covariance properties empirically validated. This is further complemented by a robust experimental setup in which DiscQuant is tested in several neural architectures (Phi-3-mini-3.8B and Meta-Llama-3.1-8B) along with quantization formats (block scaling, incoherence processing) with clear evidence that it surpasses the state-of-the-art techniques already in place like GPTQ and RTN. The results are very diversified regarding the extended coverage of benchmarks and quantization levels and therefore effectively demonstrate the applicability of DiscQuant over a wide scope and strength.

The paper is very clear in both its structured presentation of the theoretical framework and the algorithmic details. The authors do an excellent job in delineating the motivation behind DiscQuant as well as the central ideas of discrepancy theory. Figures and tables are well incorporated to help understanding in clear ways of how DiscQuant differs from other methods. While portions of the math development are tough slogging, the authors offer explanations that make intuitive sense, such as in the interpretation of results within the context of quantization grids and convex polytopes, which makes the theoretical contributions accessible to those readers with a strong technical background.

This work can be the first to provide significant influence for further work in quantization research, especially for large language models, where post-training quantization has to be very efficient for deployment on memory-constrained devices. At the same time, it frames quantization as a discrepancy problem and finds a practical rounding algorithm that achieves high compression with low loss in accuracy; thus, we contribute to new approach affecting far beyond LLMs and ranging from model deployment over mobile and embedded environments. This work further opens a route of further research into discrepancy theory in neural network quantization, inviting work that itself may eventually lead to very efficient quantization techniques. For value along the dimension of presentation, the paper is a highly significant and valuable addition to the field, advancing our understanding of effective quantization for modern, large-scale models.

**Weaknesses:**

While this paper has meaningful contribution, there are areas where it can improve with theoretical explanations of the methods, experimental validations, and practical applicability.

One area for improvement concerns the theoretical justification of why the proposed method is better than other data-dependent rounding methods, like GPTQ, in settings where the gradient covariance is not strongly low-rank. The authors assume low-rank covariance to support the theoretical bounds for DiscQuant, though it remains unclear how the method would perform when such an assumption fails to hold. Discussing some cases where the low-rank assumption is violated and offering either theoretical insights into potential limitations or proposing ways to adapt DiscQuant for such cases would strengthen the contribution. This would be an opportunity to refer to alternate bounding techniques that would account for the high-rank scenarios, or comparisons with approaches based on other assumptions.

The experimental evaluation was exhaustively done for standard models, but some additional comparisons of alternative data-dependent rounding techniques that are also effective in PTQ contexts may be very helpful. For example, newer approaches like CDQuant or AdaQuant-all seek to minimize quantization error in a data-dependent optimization manner-would make good baselines. Including the experiments that were run in those methods would actually give more background about how the comparative performance of DiscQuant goes along with an outline of relative advantages. Experiments on further tasks beyond text generation and multiple-choice questions - such as real-time inference on mobile devices or edge computing environments - should further enable these results to generalize. Extensions in this direction will unveil the flexibility and possible trade-offs of DiscQuant across different practical applications.

While clear on the whole, some aspects of the presentation of discrepancy theory in the paper could have been clearer for readers not familiar with discrepancy theory, such as explaining quantization in terms of how discrepancy theory lends itself to being quantized. As it stands now, it is not very intuitive especially to readers not familiar with convex polytope to connect the random walk the Lovett-Meka algorithm used and the rounding process in DiscQuant. The connection is necessary to be explained further in order to make the text more readable and understandable.

In conclusion, though DiscQuant strongly performs well on all benchmarks, it may extend its discussion regarding the practical impact of the deployment of DiscQuant in real-world applications. For example, since the approach is iterative, even the computational efficiency along with memory usage in comparison to lesser approaches such as RTN may be much beneficial. Examined in greater detail, the amount of increased computational expense in terms of overhead in the form of time or memory could shed light on the trade-offs between using DiscQuant and alternative approaches. Second, some comments on ease of implementation, especially in comparison to such widely used approaches as GPTQ, might be useful for practitioners testing its practical utility.

In summary, what the paper contributes is a good contribution, but it requires theoretical discussion on the possible limitations of the low-rank assumption, more experiments on a variety of baselines and tasks, clearer depiction of the role of discrepancy theory, and more practical insights into trading off deploying DiscQuant in a variety of real-world settings, thereby making the results of this paper more comprehensive, accessible, and applicable across a wider range of contexts.

**Questions:**

1. How does DiscQuant do if the gradient covariance does not have strong low rank structures? In other words, if the low rank structure of the gradient covariance does not hold, are there alternative strategies by which we could adapt or extend DiscQuant so that the discrepancies may continue to be effective? Perhaps by modulating the discrepancy-based constraints? I'd appreciate any insight into whether or not DiscQuant is extensible to other model architectures and distributions.

2. Of relevance to GPTQ and RTN, it would be interesting to see other baselines of recent data-dependent rounding techniques, such as CDQuant and AdaQuant, which optimize the quantization error. This would give a more thorough view of the possibilities and compromises with DiscQuant. Could the authors include these new baselines in future work or some ideas on how DiscQuant would theoretically compare with them?

3. The proposed method, DiscQuant, basically comprises an optimization loop. How do the computational and memory costs of DiscQuant compare to the alternatives, like RTN and GPTQ? Is the latency or memory overhead for quantization drastic? How would these effects lead to deployment challenges for applications that have specific real-time requirements or are large-scale models? A careful study of those practical tradeoffs would be useful for understanding how this approach is actually feasible in the production setting.

4. This paper relies upon the discrepancy theory as a basis for its rounding strategy, but not all readers will be as familiar with discrepancy theory as the authors. Could the authors provide additional explanation for why discrepancy theory is especially well-suited to this problem? Moreover, explanations of the concepts random walk and convex polytope would be more transparent and therefore better for readers who are unaware of these topics if they were illustrated with an example, perhaps also simplified, to bridge the gulf between such a theoretical approach and practical application.

5. In DiscQuant, the random walk approach by inspiration of the Lovett-Meka algorithm finds some feasible vertex in the polytope. Could the authors provide more intuition behind why this method is useful for quantization? For instance, does one actually need the random walk to get low generalization error, or might possibly much simpler methods for finding a vertex of the polytope work comparably? Such intuition would help clarify the design choices and could perhaps lead to avenues for simplifications.

6. The paper employs KL divergence as a metric to be minimized to reduce the gap between the original and a quantized model. Would the authors consider alternative metrics, such as MSE or other activation-based losses, for further generalizing the properties of DiscQuant? The better understanding of loss formula interactions would allow practitioners to fine-tune this method for specific applications.

7. Although DiscQuant is only tested on text-based tasks with Phi-3-mini and Meta-Llama models, the authors might highlight the scope of potential application of this work with CNNs or transformers for vision. Are there any architectural or task-specific restrictions such that the approach of DiscQuant needs to be adjusted? Further research in that would demonstrate the extensibility of DiscQuant and what needs to be changed to allow the approach for other applications.

8. The authors assume in practice that the gradient space is low rank, an assumption they verify empirically with certain architectures. Do they have additional insight or data on how DiscQuant behaves with higher-rank gradient spaces over datasets or models? Elucidation of any limits or change in performance within such settings would help one understand whether the assumptions for the method are generalizable.

---

> ### Author Response · Authors · 2024-11-22
> **Author response**
>
> **Q. What happens when the gradient covariance is not strongly low rank.**
>
> **A.** This is a great question. First of all, we are the only PTQ LLM paper which provides any kind of generalization guarantee. Doing so requires a low-rank assumption on the gradients.We can see that without low rank assumption any PTQ method will fail to generalize beyond the small calibration dataset. Suppose the gradient covariance matrix is the identity matrix $I_n$ for simplicity, i.e., it is full rank and the eigenvalues do not decay at all. In this case, any PTQ method which outputs a quantization $\hat{w}$ will incur an error of
> $$
> \mathbb{E}_s (f(\hat{w};s)-f(w;s))^2 \approx \mathbb{E}_s \langle \nabla_w f(w;s), \hat{w}-w \rangle^2 = (\hat{w}-w) [\mathbb{E}_s \nabla_w f(w;s) \nabla_w f(w;s)^T] (\hat{w} - w) = (\hat{w}-w) \Sigma( \hat{w}-w) = (\hat{w}-w)(\hat{w}-w) = \lVert \hat{w}-w \rVert ^2\propto n
> $$,
>  i.e., the quantization error $(f(\hat{w};s)-f(w;s))$ grows proportional to $\sqrt{n}$ where $n$ is the number of parameters. Or more generally if the covariance matrix has $m$ large eigenvalues, then the error will grow with $\sqrt{m}$. Therefore the decay of eigenvalues of the covariance matrix is a necessary condition for any PTQ method to work. We thank the reviewer for asking this question, we will present this argument more formally as a proposition in the updated paper.
> .
>
> **Q. Additional Experiments**
>
> **A.** CDQuant is a concurrent submission to ICLR2025, and we were unable to find their code online. We have evidence indicating GPTQ performs similar or better than AdaQuant, and therefore we believe that comparing to GPTQ is sufficient for our experiments. The “Optimal Brain Compression(OBC)” [1] paper compares to AdaQuant in their Table 4, performing similarly or better. GPTQ is the successor method to OBC focusing on computational efficiency.
>
> We ran the recent SOTA OmniQuant method in our experimental setting: Llama-3-8B-Instruct on the Wikitext, PIQA, and HellaSwag tasks. We run the OmniQuant [2] method from their official github, using their default parameters for weight-only quantization. We add the `--symmetric` argument to match the symmetric quantization grid used in our experiments. We see that our method DiscQuant beats OmniQuant on all tasks and at all bit settings.
>
>
> | Method | Wbits | Wiki$\downarrow$      | PIQA$\uparrow$ | Hella$\uparrow$ |
> |------------|---------|-----------|---------|---------|
> | -----------| 16.0   | 8.7       | 81.3   | 79.3  |
> | RTN      | 3.0     | $4.4$E$3$ | 52.4 | 29.1 |
> | GPTQ   | 3.0     | 23.2     | 66.6   | 45.8  |
> | OmniQ  | 3.0     | 315      | 55.7   | 31.9  |
> | DiscQ   | 3.0     | 15.       | 73.2   | 64.4  |
> | RTN     | 3.25    | 15.2     | 75.2  | 71.4  |
> | GPTQ   | 3.25   | 10.7     | 76.7  | 74.4  |
> | OmniQ | 3.25    | 14.6     | 74.2  | 69.1  |
> | DiscQ   | 3.25   | 10.5     | 79.1   | 75.1  |
> | RTN      | 3.5     | 12.7      | 76.7 | 73.0 |
> | GPTQ   | 3.5    | 10.4      | 77.3  | 75.1 |
> | OmniQ | 3.5     | 12.7      | 76.4  | 72.8 |
> | DiscQ   | 3.5     | 10.3      | 79.2  | 76.3 |
> | RTN      | 4.0    | 12.5      | 77.6  | 74.7 |
> | GPTQ   | 4.0    | 9.9        | 78.4  | 75.9 |
> | OmniQ  | 4.0    | 10.9      | 78.1  | 76.3 |
> | DiscQ    | 4.0   | 9.8        | 79.2  | 76.9 |
> | RTN      | 4.25  | 9.4       | 80.1   | 78.0 |
> | GPTQ   | 4.25   | 9.1       | 79.6  | 77.9 |
> | OmniQ  | 4.25   | 9.4      | 79.9   | 78.1 |
> | DiscQ   | 4.25   | 9.1       | 79.9  | 78.4  |
> | RTN      | 4.5    | 9.3       | 80.3  | 78.4  |
> | GPTQ   | 4.5    | 9.0       | 79.6   | 78.1 |
> | OmniQ | 4.5    | 9.3        | 80.1   | 78.3 |
> | DiscQ   | 4.5     | 9.1       | 80.6   | 78.7 |
>
>
> We believe that experiments “beyond text generation and multiple-choice”, or on vision models, are beyond the reasonable scope of this paper. To the best of our knowledge, other major works on PTQ (GPTQ, QuIP, OmniQuant, etc) focus on LLMs only. However, our method is general and can be used on CNNs or vision transformers.
>
> [1] Optimal Brain Compression: A Framework for Accurate Post-Training Quantization and Pruning. Elias Franta, Sidak Pal Singh, Dan Alistarh. NeurIPS 2022.
>
> [2] Shao, Wenqi, et al. "Omniquant: Omnidirectionally calibrated quantization for large language models." arXiv preprint arXiv:2308.13137 (2023).

---

> ### Author Response · Authors · 2024-11-22
> **Author response (2)**
>
> **Q. Better explain the role of discrepancy theory and relation of DiscQuant to Lovett-Meka algorithm (random walk).**
>
> **A.** Thanks for pointing this out. We will provide relevant background to readers unfamiliar with discrepancy theory. We explained the connection briefly in Section 3.1. To recall, $M$ is the matrix whose rows are per-sample gradients and we are given some fractional $y\in [0,1]^n$ which corresponds to the true weights. Our goal is to find an $x\in \\{0,1\\}^n$ such that $M(x-y)\approx 0$ (which corresponds to making the first order approximation of quantization error zero). A famous result of Lovasz et al. 1986 (cited in Section 3.1) shows that
>
> $$\min_{x \in \\{0,1\\}^n} ||M(x-y)||_{\infty} \le herdisc(M)$$
>
> where $herdisc(M)$ is the hereditary discrepancy of M defined as
>
> $herdisc(M)=\max_{S \subset [n]} \min_{z\in \\{-1,1\\}^S} ||M_S z||_{\infty}$
>
> where $M_S$ is the submatrix of $M$ formed by columns $S\subset [n]$. And there are many well-known techniques to bound herdisc(M). We didn’t go into details because of lack of space. But if the reviewers think it is useful, we can add an appendix elaborating on these connections.
>
> One important distinction concerns separating our analysis from our DiscQuant method. Discrepancy theory, including the Lovett-Meka algorithm, was used to obtain a generalization bound and obtain novel insights on the quantization problem. It just proves mathematically that low-rank gradient structure can imply existence of good quantization. However, we do not use Lovett-Meka in our actual method as it is very expensive to run. Our actual practical method DiscQuant minimizes KL divergence and a linear regularization term using a standard stochastic gradient descent approach which is very efficient. But the caveat is that we cannot prove the generalization bound holds for this practical method.
>
> **Q. Is the random walk to find a vertex of the polytope necessary for our generalization bound (in the Lovett-Meka algorithm)?**
>
> **A.** Great question! Yes, there are many ways to find a vertex of a polytope (for example maximize an arbitrary linear function on the polytope). But the random walk as done in the Lovett-Meka algorithm is necessary for our generalization bound to hold. The random walk ensures that the covariance of the vertex it finds is approximately the identity matrix and this is crucially used to prove the generalization bound. In fact, any deterministic way to pick a vertex will not generalize. But it is an interesting open question if we pick a vertex by maximizing a random linear function, then will it generalize. This is generally not true, we constructed a counter example. But with some additional assumptions, this might be true.
>
> **Q. Real world impact, compute / memory cost, deployment.**
>
> **A.** There is an important distinction between the one-time quantization cost, and the repeated inference cost, that we can make more clear. To clarify, our approach is not “iterative”, because we round all parameters at once. While our method has increased compute or memory costs compared to GPTQ and RTN, this is only occurred once during quantization. GPTQ only holds a single decoder layer in GPU memory. As explained in the top of Section 5, our method has similar memory requirements as knowledge distillation, which also requires holding two copies of the model in memory. Our method is a rounding algorithm; once the weights are rounded then there is no additional inference overhead or complexity compared to GPTQ or RTN.
>
> **Q. Alternative metrics beyond KL divergence.**
>
> **A.** Great question. Please see Table 6 in the Appendix, where try several variations of a normalized MSE between intermediate layers, and linear combinations of the KL loss and this MSE loss. We found the standard KL divergence worked best.

---

### Author Response · Authors · 2024-11-22
**Overall Comment**

We thank the reviewers for their helpful feedback. Overall, our paper is viewed as “a highly significant and valuable addition to the field, advancing our understanding of effective quantization for modern, large-scale models” (Reviewer TWTt). Reviewers appreciated the application of discrepancy theory to an under-explored area---rounding---for post-training quantization (PTQ) of LLMs. Reviewer 6GRV highlights our observation that many other LLM PTQ works “ignore the importance of the rounding step and only focus on the quantization grid”.

To the best of our knowledge, our work is the first to provide a generalization bound for PTQ, using insights from discrepancy theory to prove that low-rank gradient structure can imply the existence of good quantization. Reviewer 7uqc states our paper “provides a solid theoretical analysis to study the generalization error of quantization...It is novel to apply techniques of discrepancy theory to neural network quantization”. In addition Reviewer 6GRV states, “the assumptions necessary for the theory, seems to be well substantiated and reasonable, and authors go to reasonable length to explain and justify them, rather than hiding them”.

Building upon these insights, our algorithm DiscQuant is a practical method which uses stochastic gradient descent to minimize KL divergence and a linear regularization term to encourage quantization. Our paper conducts “extensive experiments on Phi-3-mini-3.8B and Meta-Llama-3.1-8B models across tasks and quantization levels indicat[ing] the superior performance of DiscQuant against RTN and GPTQ specifically at low bits” (Reviewer TWTt). In addition, “the proposed algorithm is agnostic to the quantization grid, which makes it quite generally applicable” (Reviewer ney3).

---

### Meta-Review · Area_Chair_xveh · 2024-12-20

**Metareview:**

Dear Authors,

Thank you for your valuable contribution to ICLR and the ML community. Your submitted paper has undergone a rigorous review process, and I have carefully read and considered the feedback provided by the reviewers.

This work proposes an approach to quantize neural networks using inspirations from discrepancy theory. The approach is evaluated on some recent language models. A theoretical result on the generalization error is also presented.

The paper received mixed review scores (3,6,3,6). Reviewers pointed out critical issues including (i) limitations of the low-rank assumption (ii) limited numerical evaluations, (iii) lack of practical insights. Thank you for providing a detailed rebuttal. However, the rebuttal was not convincing enough for three reviewers to increase their scores.

Given the current form of the paper and the reviewer discussion, I regret to inform you that I am unable to recommend the acceptance of the paper for publication at ICLR. I want to emphasize that this decision should not be viewed as a discouragement. In fact, the reviewers and I believe that your work has valuable insights and, with further development and refinement, can make a meaningful impact on the field.

I encourage you to carefully address the feedback provided by the reviewers and consider resubmitting the paper. Please use the comments and suggestions in the reviews to improve and refine your work.

Best,
AC

**Additional Comments On Reviewer Discussion:**

Reviewers TWTt and 7uqc pointed out critical issues including (i) limitations of the low-rank assumption (ii) limited numerical evaluations, (iii) lack of practical insights. The authors provided a detailed rebuttal, however, the rebuttal was not convincing enough for three reviewers to increase their scores.

---

### Decision · Program_Chairs · 2025-01-22

Reject